# Efficient neural representation in the cognitive neuroscience domain: Manifold Capacity in One-vs-rest Recognition Limit

## Abstract

The structure in neural representations as manifolds has become a popular approach to study information encoding in neural populations. One particular interest is the connection between object recognition capability and the separability of neural representations for different objects, often called "object manifolds." In learning theory, separability has been studied under the notion of storage capacity, which refers to the number of patterns encoded in a feature dimension. Chung et al. (2018) extended the notion of capacity from discrete points to manifolds, where manifold capacity refers to the maximum number of object manifolds that can be linearly separated with high probability given random assignment of labels. Despite the use of manifold capacity in analyzing artificial neural networks (ANNs), its application to neuroscience has been limited. Due to the limited number of "features", such as neurons, available in neural experiments, manifold capacity cannot be verified empirically, unlike in ANNs. Additionally, the usage of random label assignment, while common in learning theory, is of limited relevance to the definition of object recognition tasks in cognitive science. To overcome these limits, we present the Sparse Replica Manifold analysis to study object recognition. Sparse manifold capacity measures how many object manifolds can be separated under one versus the rest classification, a form of task widely used in both in cognitive neuroscience experiments and machine learning applications. We demonstrate the application of sparse manifold capacity allows analysis of a wider class of neural data - in particular, neural data that has a limited number of neurons with empirical measurements. Furthermore, sparse manifold capacity requires less computations to evaluate underlying geometries and enables a connection to a measure of dimension, the participation ratio. We analyze the relationship between capacity and dimension, and demonstrate that both manifold intrinsic dimension and the ambient space dimension play a role in capacity.

## 1 Introduction

The approach to study neural populations as manifolds and their geometry has become a popular method to uncover important structural properties in neural encoding and understand the mechanisms behind the ventral stream, the motor cortex, and cognition (Kriegeskorte & Kievit, 2013)(Sengupta et al., 2018) (Gallego et al., 2017) (Sohn et al., 2019)(Ebitz & Hayden, 2021)(Kriegeskorte & Wei, 2021) (Chung & Abbott, 2021). In the ventral stream, the invariant ability for humans and animals to recognize an object despite changes in pose, position, and orientation has motivated a definition of object manifold as the underlying representation of neural responses to a distinct object class. A long-standing hypothesis in visual neuroscience posits that the visual cortex untangles these object manifolds for invariant object recognition (Dicarlo & Cox, 2007), relating object recognition to the separation of manifolds by some linear hyperplane.

There is a well developed theory of linear separability given by Gardner (1988) that studies the separation of points by a perceptron. The theory quantifies a capacity load that describes the maximum number of points that can be linearly separated given a random dichotomy (a random assignment of binary labels to the manifolds). The capacity load also encodes the number of points stored per feature dimension required to have linear separability. This theory of separation, however, does not connect to the geometries of the underlying representations.

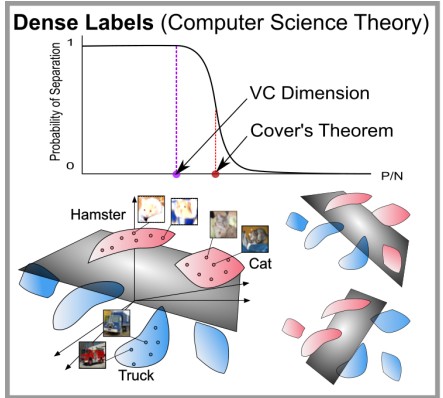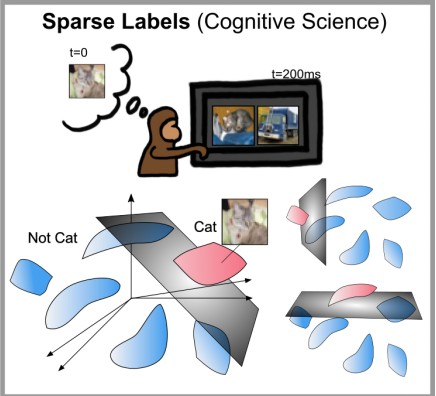

Figure 1: **Dense vs Sparse labels** (*left*) Computer science theories such as VC dimension and Cover's Theorem are interested in capacity under all random dichotomies. Under this regime, manifolds are densely labeled where more than one manifold has a positive label. (*right*) Conversely, neuroscience studies cognitive tasks such as object recognition, where one-vs-rest dichotomies are more relevant. Under this regime, manifolds are sparsely labeled where only one manifold is assigned a positive labeling at a time. In each dichotomy, (red) is the positive label 1 and (blue) is the negative label -1.

Chung et al. (2018) extended the notion of capacity from points to manifolds and related capacity to manifold geometric properties. Manifold capacity, defined as the maximum number of object manifolds that can be linearly separated given a random dichotomy, has been used to show how classification emerges along layers of deep neural network (DNN) undertaking visual object recognition Cohen et al. (2020) and speech recognition (Stephenson et al., 2019). Furthermore, it has been shown that geometric properties of the manifold, defined as manifold radius and dimension, decrease in magnitude while capacity increases. This observation suggests feature representations with a low intrinsic dimension allows invariance and robustness in classification.

Despite the use of manifold capacity in analyzing DNN, its applicability to neuroscience has been limited due to the way manifold capacity is defined. The current theory following Gardner's framework considers all random dichotomies, similar to other theoretical computer science studies. VC dimension in learning theory returns the largest number of points in a set such that every dichotomy of the set can be learned by a given hypothesis function (Vapnik & Chervonenkis, 2015; Abu-Mostafa et al., 2012). Cover's theorem returns the number of linearly separable sets given $P$ points and a $D$ dimensional space (Cover, 1965). The regime of object recognition for a classification model and for a monkey performing a delayed matching Majaj et al. (2015b) or oddity task, however, is equivalent to the separation of manifolds on a one-vs-rest basis (Figure 1). In other words, the only relevant dichotomies are those where only one manifold has a positive labeling. This relates to the notion of sparse labeling in Chung et al. (2018).

Sparse labeling also overcome the technical restrictions of using manifold capacity to analyze biological neurons like previous works have for artificial neurons. The current manifold capacity theory for random dichotomies falls outside the regime of most available neural datasets, namely, data with limited number of simultaneously recorded neurons (Gao et al., 2017). Hence, as modern large scaled probing techniques improve and become publicly available (Jia et al., 2019; Steinmetz et al., 2021), analyzing current data requires further innovations in theoretical and analysis framework. Under sparse labeling, capacity is greater than in the traditional regime (Chung et al., 2018; Gardner, 1988). It follows that, under the sparse label regime, we can verify capacity in datasets with fewer number of features, or neurons, which was not previously possible (Froudarakis et al., 2021). Thus, the sparse labeling regime allows us to apply the theory of manifold capacity to real neural data and use capacity as a measure of recognition and similarity between DNN and the biological brain.

In this paper, we extend the work presented in Chung et al. (2018) to analyze neural data by estimating manifold capacity in the one-vs-rest recognition limit. We define sparse manifold

capacity as the ratio of $P/N$ where $P$ is the number of manifolds in a given system, and $N$ is the degree of freedom required such that the probability of linearly separating a one-vs-rest dichotomy is 0.5. Our contribution is as follow:

- We present the Sparse Replica Manifold analysis for estimating sparse manifold capacity in the one-vs-rest recognition limit. Our analysis overcomes the limitations in Chung et al. (2018) by taking into account of correlation between manifolds and the effects of a heterogeneous mixture of manifold geometries, thereby making the analysis applicable to real complex data.[1]

- We show that the application of sparse manifold capacity allows analysis of a wider class of neural data that has a limited number of neurons with empirical measurements. We demonstrate, for the first time to our knowledge, the match between the theoretical and empirical manifold capacity in real neural data, and suggest capacity as a reliable measurement of linear separability in the biological brain.

- We show that the manifold geometries under the sparse label regime is faster to compute than in the classical regime, allowing efficient analyses via manifold geometries.

- We explicitly illustrate the effects of ambient dimension and manifold intrinsic dimension on sparse linear separability. In particular, we show that better linear separability is related to both high ambient dimension and low manifold intrinsic dimension, filling in the gap of knowledge on the role of high and low dimensional representations in neural encoding.

## 2 Review of Mean Field Theory Manifold Analysis

We will first summarize the Mean Field Theory Manifold Analysis (MFTMA) for object manifolds first introduced in Chung et al. (2018). This framework extends the calculation of perceptron capacity of discrete points (Gardner, 1988) to capacity of manifolds. Consider a system of $P$ manifolds in a $N$-dimensional ambient space, where each manifold has an affine dimension of $D$. We define the capacity load $\alpha_c = P/N_c$ where $N_c$ is the critical number of feature dimensions required such that 0.5 of all manifold dichotomies are separable by a hyperplane. $\alpha_c$ can also be interpreted as the number of categories encoded per feature dimension. Chung et al. (2018) calculated capacity as an average of $\alpha_M$, for each manifold $M$. To determine $\alpha_M$, equation 1[2] involves finding manifold anchor points, $\tilde{s}(\vec{t}, t_0)$, uniquely determined by Gaussian vectors $T \in \mathbb{R}^{D+1}$, whose components $T_i \backsim N(0, 1)$. For the rest of this paper, we denote $T = (\vec{t}, t_0)$ ($\vec{t} \in \mathbb{R}^D, t_0 \in \mathbb{R}$). Given some dichotomy and orientation of the manifolds, the anchor point is a point on the manifold or its convex hull that contributes to the separating linear hyperplane. The notion of anchor point emerges from the solution to the KKT interpretation of the mean field equations for capacity, discussed in depth in Chung et al. (2018) (see also appendix A.8). Similar to support vectors in a Support Vector Machine, these anchor points change based on the location and labels of other manifolds. The Gaussian vectors represent the variability in orientations, locations, and labels of other manifolds.

$$\alpha_M^{-1} = \left\langle \frac{[-t_0 - \vec{t} \cdot \tilde{s}(\vec{t}, t_0)]^2}{(\|\tilde{s}(\vec{t}, t_0)\|^2 + 1)^2} \right\rangle_{\vec{t}, t_0} \tag{1}$$

MFTMA links the connection between this capacity load with the manifold geometric properties. In particular, Chung et al. (2018) defined for each manifold, the manifold anchor radius ($R_M$) squared as the mean squared length of all $\tilde{s}(\vec{t}, t_0)$, i.e. $R_M^2 = \langle \|\tilde{s}(\vec{t}, t_0)\|^2 \rangle_{\vec{t}, t_0}$. In addition, the manifold anchor dimension ($D_M$) measures the mean angular spread between the unit vector of an anchor point ($\hat{s}$) and the Gaussian vector that determines the anchor point, i.e. $D_M = \langle (\vec{t} \cdot \hat{s}(\vec{t}, t_0))^2 \rangle_{\vec{t}, t_0}$ Chung et al. (2018) also showed that one can estimate $\alpha_M$ using manifold anchor radius and dimension by $\alpha_{Ball}(\kappa, R_M, D_M)$ with a margin of $\kappa$,

$$\alpha_{Ball}^{-1}(\kappa, R, D) = \int_0^\infty dt \chi_D(t) \cdot \left[ \int_{\kappa - tR^{-1}}^{\kappa + tR} Dt_0 \frac{(-t_0 + tR + \kappa)^2}{(1 + R^2)} + \int_{-\infty}^{\kappa - tR^{-1}} Dt_0 \left( (t_0 - \kappa)^2 + t^2 \right) \right] \tag{2}$$

where $Dt$ is the Gaussian measure and $\chi_D(t) = \frac{2^{1 - \frac{D}{2}}}{\Gamma(\frac{D}{2})} t^{D-1} e^{-\frac{1}{2}t^2}, \quad t \geq 0$.

---

[1] Upon publication, the Sparse Replica Manifold Analysis will be made publicly available for the use of the neuroscience and machine learning community to analyze artificial neural networks and biological data.

[2] $\langle \cdot \rangle$ denotes taking the average in this paper.

## 2.1 GAUSSIAN RADIUS AND GAUSSIAN DIMENSION

Shown in Chung et al. (2018), for small manifold sizes, the anchor point $(\tilde{s})$ depends only on $\vec{t}$. This motivates a definition of manifold radius similar to the Gaussian mean width. Chung et al. (2018) additionally defined the Gaussian radius $(R_g)$ and Gaussian dimension $(D_g)$ as

$$R_g^2 = \langle \|\tilde{s_g}(\vec{t})\|^2 \rangle_{\vec{t}}, \, D_g = \langle (\vec{t} \cdot \hat{s_g}(\vec{t}))^2 \rangle_{\vec{t}} \tag{3}$$

where $\tilde{s_g}(\vec{t})$ is the anchor point determined by $\vec{t}$, and $\hat{s_g}(\vec{t})$ is its unit vector. These measures play important roles when considering classification tasks with sparse labels (see next section).

## 3 MANIFOLD CAPACITY FOR ONE-VS-REST OBJECT RECOGNITION

The MFTMA estimates the capacity load when $0.5$ of all $2^P$ dichotomies in system of $P$ manifolds are separable. In neuroscience and learning models, however, a more practical usage of this theory is to estimate capacity where manifolds are separated on a one-vs-rest basis. This falls under the notion of sparse labeling, in contrast to dense labeling. In other words, we want to consider only the $P$ dichotomies, in each which only one manifold has a positive labeling. Assuming that each manifold represents a class of object, estimating capacity on a one-vs-rest basis is equivalent to estimating the capacity for a system undertaking an object recognition task.

In the following sections, we present the Sparse Replica Manifold Analysis that computes manifold in the one-vs-rest recognition limit, also called sparse manifold capacity. Sparse manifold capacity was previously considered in Chung et al. (2018) in the context of random uncorrelated manifolds with identical geometries (i.e. every manifold has the same manifold radius and dimension). We extend this work by taking into account of heterogeneous geometries and correlations between manifolds. In the following sections, we first review the theory for sparse capacity considered in previous work. Then, we provide the theorem for calculating sparse manifold capacity in the general case with a mixture of different manifold geometries. Lastly, we discuss the case where manifold center correlation are taken into account.

### 3.1 SPARSE MANIFOLD CAPACITY

Define $f$, the sparsity parameter, as the fraction of positively labeled manifolds in a dichotomy. Then, $f = 1/P$ corresponds to the one-vs-rest dichotomies. Under the regime of sparse labeling, capacity depends on maximizing a bias parameter, where the hyperplane is not constrained to pass through the origin. This bias parameter has a positive contribution to the margin for the positively labeled manifold and a negative contribution for the negatively labeled manifolds. The resulting effect of the bias parameter, shown in Chung et al. (2018), puts the manifolds in regime where $t_0$ becomes negligible to the anchor point of a contributing manifold. These observations allow us to use equation 4 to estimate the sparse manifold capacity assuming every manifold behaves as a ball with the same Gaussian radius $(R_g)$ and Dimension $(D_g)$ (Chung et al., 2018).

$$\alpha_{c,HOMOG}(\kappa, R_g, D_g) = \max_b [f \cdot \alpha_{Ball}^{-1}(\kappa + b, R_g, D_g) + (1 - f) \cdot \alpha_{Ball}^{-1}(\kappa - b, R_g, D_g)]^{-1} \tag{4}$$

#### 3.1.1 SPARSE MANIFOLD CAPACITY WITH MIXTURES OF SHAPES

In this work, to account for the different Gaussian Radii and Dimensions for each manifold in a heterogeneous system, we derive and employ Theorem 1. The derivation of this theorem is provided in the appendix A.1, and a short version is given in the next section. For each manifold $i = 1, 2, ...P$, we calculate $\alpha_{M^i}$ from Equation 5, where $\vec{R}$ is the array of all manifold Gaussian Radii in the system, and $\vec{D}$ is the array of all manifold Gaussian dimensions in the system. The sparse manifold capacity is the average of all $\alpha_{M^i}$.

**Theorem 1.** *Given $P$ manifolds. Let $\vec{D_g} = [D_{g1}, ...D_{gp}]$ be the array of their corresponding Gaussian dimensions. And let $\vec{R_g} = [R_{g1}, ...R_{gp}]$ be the array of their corresponding Gaussian radii. Then sparse manifold capacity, $\alpha_{c,SPARSE}$, is computed as $\langle \alpha_{M^i} \rangle_{i=\{1,2...P\}}$ where*

$$\alpha_{M^i}(\kappa, \vec{R_g}, \vec{D_g}) = \max_b [1/P \cdot \alpha_{Ball}^{-1}(\kappa + b, R_{gi}, D_{gi}) + \sum_{j \neq i} 1/P \cdot \alpha_{Ball}^{-1}(\kappa - b, R_{gj}, D_{gj})]^{-1} \tag{5}$$

### 3.1.2 DERIVATION TO THEOREM 1

Given $\vec{D}_g$ and $\vec{R}_g$, by (Chung et al., 2018), the capacity of such ensemble of manifolds is the average of $\alpha_M$, computed for each manifold by equation 1 or estimated as $\alpha_{BALL}(\kappa, R, D)$ (equation 2). Sparse capacity for a system of manifolds with identical shapes, is computed by equation 4. Observe that equation 4 is averaging across $\alpha_{BALL}$'s, each with inputs from a manifold with either positive labeling or negative labeling indicated by $+b$ and $-b$. Since all $R$ and $D$ are the same in a homogeneous system, equation 4 sufficiently accounts for all possible assignments of positive and negative labeling. In the case of heterogeneous geometries, it follows that we need to consider all possible dichotomies given by the sparse parameter. In order to conserve equation 4, averaging across all such dichotomies suffices. For the one-vs-rest regime discussed in this paper, there are only $P$ dichotomies to consider, within each only one manifold is positively labeled. Thus, for each manifold $i$, compute equation 5. Then, averaging across all $\alpha_{M^i}$, i.e. across all relevant dichotomies, gives us the sparse manifold capacity.

### 3.1.3 SPARSE MANIFOLD CAPACITY WITH CENTER CORRELATION

MFTMA assumes that the manifold centers of the system are randomly related. Realistic data, however, tend to exhibit correlation. Cohen et al. (2020) took center correlation into account by projecting the manifolds into a space where they will have low-correlated centers. This low correlation space is determined via the Euclidean centroids of each manifolds. We follow a similar approach, but instead of using the Euclidean centroids of each manifold to recover the space of low correlations, our approach for computing sparse manifold capacity with center correlation uses **categorical local linear differences (CLLD)**, which are sampled differences between the local linear centers of the manifolds.

Suppose manifold $M \in \mathbb{R}^D$. We define a manifold's **local linear centroid** as the D-dimensional projection of its Euclidean centroid in the LLE space. The LLE space is the lower $d \ll D$ dimensional space that resulted from the local linear embedding (LLE) (Roweis & Saul, 2000) of a manifold[3]. We describe the algorithm to recover the appropriate low correlations space using CLLD and calculate sparse manifold capacity in figure 2. In appendix A.4, we show our method is better than the technique in Cohen et al. (2020) for estimating sparse manifold capacity with correlations.

**Computing sparse manifold capacity with center correlation using CLLD.**
1: Use LLE to project each manifold into a low dimensional space
2: For each manifold, compute its Euclidean center in the new space and project the center back to the original, high dimensional space to obtain local linear centroids.
3: Sample CLLD by sampling differences between local linear centroids.
4: Use CLLD to recover the space of low center correlation via the technique in Cohen et al. (2020).
5: Project each manifold into the low center correlation space and compute capacity via Theorem 1

Figure 2: Pseudocode for sparse manifold capacity with correlation. See appendix A.2 for details.

## 4 RESULTS

By relaxing the homogeneous and random correlations assumptions in Chung et al. (2018), our method estimates manifold capacity for real data in the one-vs-rest recognition limit, the relevant domain for machine learning and neuroscience. While previous works (Cohen et al., 2020; Stephenson et al., 2019) estimate manifold capacity in ANNs under the dense label regime, we demonstrate for the first time the application of manifold capacity in the neuroscience domain, enabled by sparse labeling. We use our Sparse Replica Manifold Analysis to measure the sparse manifold capacity of deep neural network manifolds and neural manifolds collected from the ventral streams of primates presented with visual stimulus in experiments done by Majaj et al. (2015a) and Freeman et al. (2013). Details of these datasets are provided in the appendix A.3. We show that sparse manifold capacity requires fewer number of features, ideal for neural data where limited number (e.g., order of hundreds) of neurons are recorded. With this, we demonstrate the match between the theoretical and empirical manifold capacity in real neural data for the first time (to our knowledge). We then study the manifold

---

[3]For the experiments in this paper, we chose d=2 to minimize the reconstruction error of LLE projection

geometries in relation to sparse manifold capacity and show explicitly that smaller manifold intrinsic dimension is desirable for larger capacity and better linear separability.

## 4.1 SPARSE MANIFOLD CAPACITY THEORY PREDICTS NUMERICS IN REAL DATA

In this section, we show how well theorem 1 predicts the sparse manifold capacity. Figure 3 demonstrates the match between the theoretical sparse manifold capacity (determined by our theorem) and simulated sparse manifold capacity (the ground truth determined empirically) for responses in layers of a CIFAR-100 trained VGG-16 and brain regions of Macaque monkeys to given stimuli. The ground truth sparse manifold capacity is determined by interpolating the critical number of features ($N_c$) over which the probability a one-vs-rest dichotomy is linearly separable drops from 1 to 0. In practice, we use a bisection search for $N_c$ where the probability is nearly 0.5. The linear separability of each dichotomy is determined by finding a consistent SVM model. The full algorithm to determine simulated capacity is given in the appendix A.6. Interestingly, it appears that the neural data in figure 3 is better estimated by theorem 1. A possible explanation is that neural data have existing neural noise that smooths out the manifold geometries and makes them more amenable to our analysis.

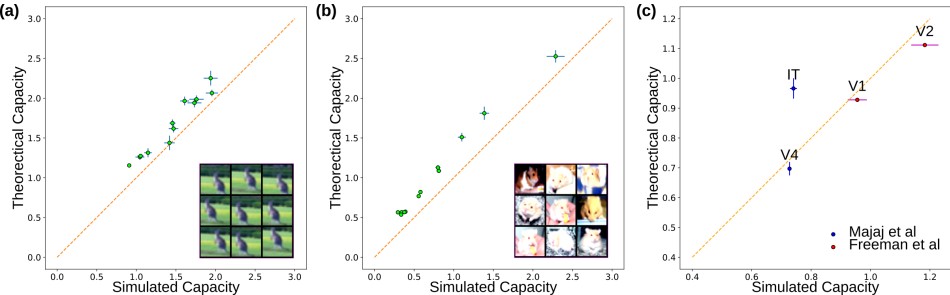

Figure 3: **Theoretical vs simulated sparse manifold capacity (a)** on smooth manifolds in layers of a VGG16 **(b)** on pointcloud manifolds in layers of a VGG16 **(c)** on neural data from the ventral stream of Macaque monkeys engaged in a classification task. Each dot is a layer or a brain region.

## 4.2 BROADER APPLICATION OF MANIFOLD ANALYSIS ENABLED BY SPARSITY

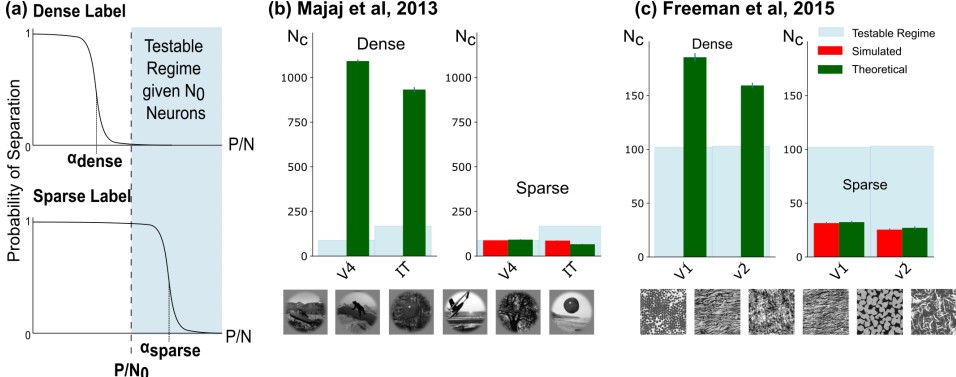

Figure 4: **(a)** Sparse Replica Manifold Analysis enables simulation-theory validation in real neural data with limited number of the neurons. Given $N_0$ neurons in the data, capacity under dense labels tends to be outside the testable regime (*top*). Conversely, under sparse labels, the theoretical capacity can be validated empirically (*bottom*). In **(b)** and **(c)**, we demonstrate this phenomenon. We estimate critical number of features $N_c = P/\alpha_c$ from the neural data in V4 and IT regions (Majaj et al., 2015a) and the V1 and V2 regions (Freeman et al., 2013) under dense labels (*left*) and sparse labels (*right*). The testable regime *(highlighted blue)* depends on the number of neurons/features available. Sample images of each dataset are shown on the bottom of their respective panel. Observe that under dense labeling, there is no simulated data available for validation.

Sparse labeling allows analysis of capacity on a wider class of neural data that has a limited number of neurons measured. Recall that capacity is the ratio of $P/N_c$ (refer to section 2). Figure 4a (*top*) demonstrates that under dense labels and limited number of neurons/features, $N_0$, capacity often cannot be validated empirically using the bisection search because $N_c \gg N_0$. On the other hand, the verification for capacity under sparse labeling requires less number of features (Figure 4a *bottom*). For the Majaj dataset (Figure 4b), the estimation for $N_c$ under dense labeling requires almost a thousand neurons in the IT region where only 168 neurons were recorded. Under sparse labeling, the estimation is reduced to 65 neurons. Similarly, for the Freeman dataset (Figure 4c), dense labeling estimates 185 and 159 neurons for the V1 and V2 regions while sparse labeling estimates just 32 and 27 neurons, well within the number of neurons available.

## 4.3 COMPUTATIONAL EFFICIENCY

We observe that the computational time for the relevant manifold geometries is faster under sparse labeling compared to dense labeling by an order of magnitude. Table 1 compares the CPU time for computing manifold radius and dimension under dense labeling and sparse labeling for layers of the VGG-16. We note that the total sequential computation time for sparse manifold capacity is dominated by the optimization of the $b$ parameter, and it is typically not faster compared to capacity under dense labels (see appendix A.5). Nonetheless, sparse manifold capacity enables the use of Gaussian radius and Dimension as a convenient measure of manifold geometry during object recognition.

Table 1: Sparse Replica Manifold Analysis Computation Time (seconds): Manifold Radius and Dimension computed sequentially for 40 manifolds.

| VGG-16 Layers | Input | ReLU-4 | ReLU-7 | ReLU-14 | ReLU-21 | ReLU-28 | ReLU-37 |
|---|---|---|---|---|---|---|---|
| Sparse Labels | **1.01** | **1.16** | **1.23** | **1.14** | **1.15** | **0.76** | **1.07** |
| Dense Labels | 51.68 | 50.00 | 50.30 | 48.53 | 46.34 | 45.76 | 45.10 |

## 4.4 MANIFOLD GEOMETRY AND SPARSE CAPACITY IN ARTIFICIAL AND NEURAL DATA

In this section, we study the behavior between capacity, manifold Gaussian radius and Dimension, along with the manifold effective dimension and radius. Imposing hierarchy on layers of the DNN and the ventral stream, figure 5 shows that capacity generally increases while average manifold Gaussian dimension ($D_g$) and Gaussian radius decrease ($R_g$). This follows the similar observations made in Cohen et al. (2020) and suggests that compressing the manifold allows better manifold linear separability.

Figure 5 also compares the Gaussian geometries we have introduced to the notion of effective dimension and effective radius. Effective dimension and effective radius correspond to measuring the manifold dimension and radius on an ellipsoid imposed on the manifold, whereas the Gaussian geometries are measured on the convex hull of each manifold. Under sparse labeling, the effective dimension is also known as the participation ratio in most literature (Chung et al., 2018; Gao et al., 2017; Litwin-Kumar et al., 2017; Elmoznino & Bonner, 2022; Jozwik et al., 2019; Sorscher et al., 2021). In figure 5(a), mean effective dimension and radius behave similarly to mean Gaussian dimension and radius for layers of the DNN. Hence, the manifold intrinsic dimension, measured by the participation ratio of each manifold, also decreases as capacity increases. We note that figure 5(b-c)(ii,iv) shows that mean effective dimension and mean Gaussian dimension do not decrease across the hierarchy imposed on the biological ventral stream as evidently as mean effective radius and mean Gaussian radius. Instead, the manifold Gaussian dimension and effective dimension appear to exhibit the opposite behavior relative to the artificial network. These suggestive differences between the artificial and the biological brain points toward using manifold geometries as a metric to finding and fitting an artificial brain model.

## 4.5 ROLE OF DIFFERENT DIMENSIONS IN CAPACITY AND LINEAR SEPARABILITY

We have shown that capacity increases while manifold Gaussian radius and Dimension decrease. This suggests that compressing the manifolds and reducing their dimensions allow better separation. This may seem to contradict the intuitive belief in deep learning that higher dimensions allow better

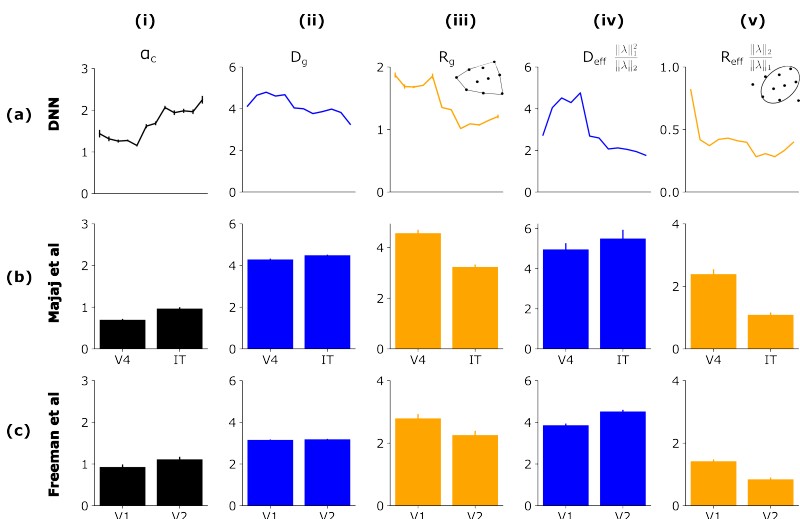

Figure 5: This figure plots **(i)** capacity, **(ii)** mean manifold Gaussian dimension, **(iii)** mean manifold radius, **(iv)** mean manifold effective dimension , and **(v)** mean manifold effective radius for smooth manifolds in ReLU layers of the **(a)** VGG16, **(b)** neural data in the V4 and IT regions, and **(c)** in the V1 and V2 regions. Effective dimension is computed as $\|\lambda\|_1^2/\|\lambda\|_2$ where $\lambda$ is the eigenspectrum of the covariance matrix of manifold. Effective radius is computed as $\|\lambda\|_2/\|\lambda\|_1$.

classification. In Figure 6, we give experimental evidence how linear separability is related to both lower manifold intrinsic dimension ($D$) and larger ambient dimension ($N$). Figure 6a summarizes our point: ambient dimension allows more expressivity for a separating hyperplane, but when the manifold intrinsic dimension is high, the manifolds may not be separable.

For these experiments, we examine the interaction between manifold intrinsic dimension (as gaussian mean width and effective dimension), ambient dimension ($N$), the sparse manifold capacity, and the margin of linear separability in the SVM (i.e. the distance between the anchor point and separating hyperplane). We sample 20 manifolds, consisting of 1000 examples in the CIFAR100 dataset. We vary the intrinsic dimension by projecting each manifold to a number of their PCA components. We vary the ambient dimension by random projection. In figure 6c,d,f,g, we illustrate that at a fixed ambient dimension, increasing intrinsic dimension leads to decreasing capacity and margin. On the other hand, at a fixed manifold intrinsic dimension, increasing ambient dimension increases capacity and margin (Figure 6b,e). Hence, even though higher ambient dimension may enable better linear separability, the intrinsic dimension still plays a role.

## 5 DISCUSSION

We devised the Sparse Replica Manifold analysis for estimating sparse manifold capacity that is applicable to neuroscience experiments. As Gardner's original perceptron capacity demonstrated that the sparse labeling regime increases capacity (Gardner, 1988), our new theory allows applications of the replica mean field theory analysis on neural data with few feature dimensions or neurons recorded. Thus, we can begin analyzing the dynamics of the neural system where only sufficient samples of neurons were drawn (Gao et al., 2017). Our theory can estimate sparse manifold capacity in layers of a deep neural network and in real neural data of the ventral stream (Figures 3,4). We are able to estimate the number of features required to separate a one-vs-rest dichotomy with 0.5 success rate within the regime of available number of features.

In contrast to the capacity used in Cohen et al. (2020), sparse manifold capacity depends only on the Gaussian radius and Dimension of each manifold, which is faster to compute than in the classical regime (Table 1). We note, however, that the time takes to compute sparse manifold capacity is longer, attributed to the optimization of the bias parameter, which warrants future studies. To account for correlation between manifolds, we also differ from Cohen et al. (2020) by

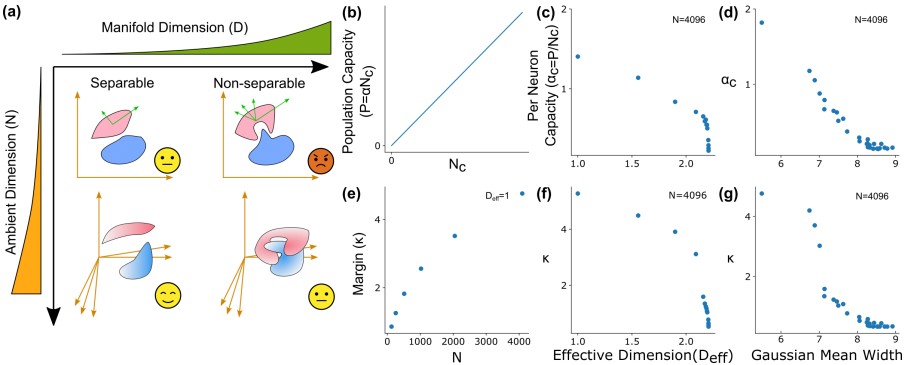

Figure 6: **(a)** illustrates the relationship between ambient dimension $N$ (shown as *orange* axes), manifold intrinsic dimension $D$ (shown as *green* axes), and linear separability. Manifolds are most easily separable in the regime of high ambient dimension and low manifold dimension. **(b)** shows that with a fixed $\alpha$, increasing $N_c$ (critical number of features) will increase the population capacity, which is the number of separable manifolds, $P$. Recall that we only need $N \gg N_c$. Otherwise $\alpha$ is not dependent on $N$. (b) holds true as long as $N \gg N_c$. Refer to section 2 for notations. **(c)**-**(g)** are empirical results on sparse manifold capacity and the mean margin achieved by a SVM when linear separability is plausible. **(c)**-**(d)** demonstrate that the per neuron capacity, which is the sparse manifold capacity, decreases as effective dimension (participation ratio) and Gaussian mean width ($R_g\sqrt{D_g}$) increase respectively. **(e)** illustrates that the margin of the SVM increases as the ambient dimension increases. We increase ambient dimension using random projections. **(f)**-**(g)** on the other hand demonstrate that margin decreases as effective dimension and the Gaussian mean width increases.

using categorical local linear differences (CLLD) to compute a space of low correlation. Thus, we introduce another method to deal with the center correlation for manifold analysis.

Generally, increasing capacity corresponds to decreasing manifold radius and dimension (Figure 5), and figure 6 shows that linear separability is associated with both low manifold dimension and high ambient dimension. The focus on the effective ambient dimension has been used in the literature to describe the encoding capability and generalization performance of neural networks (Elmoznino & Bonner, 2022; Jozwik et al., 2019) with claims that higher ambient dimensional measures benefit performance (Elmoznino & Bonner, 2022). Our work studies the manifold intrinsic dimension and illustrates the two types of dimension may behave inversely, filling in the gap of knowledge in the argument between the role of high and low dimensional representations in neural encoding.

Sparse manifold capacity can be related to other work involving sparseness in neural encoding. Previous work associates sparse neural connectivity with better readout performance (Babadi & Sompolinsky, 2014; Litwin-Kumar et al., 2017). These ideas of sparseness are associated with the number of active neurons, where in our work could relate to the number of critical features needed to achieve the given capacity of manifolds, which extends to the representation's efficiency. Sparse manifold capacity is used for the regime of one-vs-rest class discrimination, where higher sparse manifold capacity is associated with improvement in manifold separability, equivalent to better readout performance.

Our theory opens many future directions in neuroscience and machine learning. Sparse manifold capacity broadens the operating regime of the replica manifold analysis framework to lower ambient dimensions, enabling analysis of both artificial and biological neural networks occupying low dimensional space. Among other things, we hope this work will (a) enable broader accessibility of the present manifold analysis approach to a wider experimental neuroscience community; (b) motivate the usage of present geometric approach to measure and train a broader size range of machine learning models; and (c) inspire future studies in using geometric approaches like ours as method to compare representations between biological neural data and artificial neural network models of the brain, by providing theoretically grounded metrics in representation geometry and task performance.

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

## A  APPENDIX

### A.1  DERIVATION TO THEOREM 1

Suppose we have a heterogeneous system of $P$ manifolds. Let $\vec{D}_g = [D_{g1}, ... D_{gp}]$ be the array of their corresponding Gaussian dimensions. And let $\vec{R}_g = [R_{g1}, ... R_{gp}]$ be the array of their corresponding

Gaussian radii. By Chung et al. (2018), the capacity of such ensemble of manifolds is the average of $\alpha_M$, computed for each manifold by equation 1 or estimated as $\alpha_{BALL}(\kappa, R, D)$ (equation 2) where $R$ and $D$ are the manifold's respective measure of dimension and radius. Furthermore, by Chung et al. (2018), sparse capacity for a homogeneous system of manifolds (i.e.assuming all manifolds have the same Gaussian radius and Gaussian dimension), is computed as

$$\alpha_{HOMOG} = \max_b [f \cdot \alpha_{Ball}^{-1}(\kappa + b, R_g, D_g) + (1 - f) \cdot \alpha_{Ball}^{-1}(\kappa - b, R_g, D_g)]^{-1} \qquad (\star)$$

where $f$ is the sparse parameter (i.e. the fraction of positively labeled manifolds), and $b$ is the bias parameter that has positive contributions to positively labeled manifolds and negative contributions to negatively label manifolds. Observe that equation $(\star)$ is thus averaging across $\alpha_{BALL}$'s, each with inputs from a manifold with either positive labeling or negative labeling indicated by $+b$ and $-b$. Since all $R$ and $D$ are the same in a homogeneous system, equation $(\star)$ sufficiently accounts for all possible assignments of positive and negative labeling.

In the case of heterogeneous geometries, it follows that we need to consider all possible dichotomies given by the sparse parameter. In order to conserve equation $(\star)$, averaging across all such dichotomies suffices. For the one-vs-rest regime discussed in this paper, there are only $P$ dichotomies to consider, within each only one manifold is positively labeled. Thus, for each manifold $i$, compute

$$\alpha_{M^i}(\kappa, \vec{R_g}, \vec{D_g}) = \max_b [1/P \cdot \alpha_{Ball}^{-1}(\kappa + b, R_{gi}, D_{gi}) + \sum_{j \neq i} 1/P \cdot \alpha_{Ball}^{-1}(\kappa - b, R_{gj}, D_{gj})]^{-1}$$

Then, averaging across all $\alpha_{M^i}$, i.e. across all relevant dichotomies, gives us the sparse manifold capacity. Observe that this general theorem does not deviate from the case of a homogeneous system. In the case of $f = 1/P$ and we have a homogeneous system,

$$\alpha_{HOMOG} = \alpha_{M^i} \underset{i \neq j}{=} \alpha_{M^j} = < \alpha_{M^i} >$$

## A.2 Computing sparse manifold capacity via CLLD

Algorithm 1 gives the detailed pseudocode to calculate sparse manifold capacity with center correlation. For the experiments in this paper, the number of components chosen for local linear embedding of each manifold was heuristically chosen as the default value of 2 to minimize the reconstruction error of the dimensional reduction. See section 3.1.3 for definitions.

### A.2.1 Algorithms for Localizing Local Linear Centroid

Discovering the local linear centroid involves projecting each manifold low dimensional center to the high dimensional space. Since this problem involves solving an overdetermined system, we use algorithms to localize for the local linear centroid. Here we describe two of these algorithms. Results of the paper use the Nearest Neighbor algorithm.

The Nearest Neighbor algorithm tracks the $K$ nearest points on the manifold closest to the Euclidean centroid in the LLE space. The D-dimensional local linear center is the average of these $K$ points in the higher D-dimensional space. On the other hand, the Weighted Average algorithm determines a weighted average in the D-dimensional space based on the distance of each example of the manifold from the Euclidean centroid in the LLE space. Points on the manifold closer to the LLE space Euclidean centroid will have a greater weight. The algorithms to determine local linear centroids given a manifold and its LLE equivalent are provided in Algorithms 2 and 3

## A.3 Datasets

We used three datasets for our experimental analysis. In figure 3, capacity is measured across layers of a trained VGG-16 and brain regions of Macaque monkeys. For the DNN, we extract the activation from each layer to form manifolds corresponding to a given input. Responses to inputs of the same class form an object manifold. A pointcloud manifold consists of distinct examples from the same class. A smooth manifold is generated by affine transformations (translation) of an image belonging to a distinct class. Neural data were taken from Majaj et al. (2015a) and Freeman et al. (2013) where images were shown to Macaque monkeys, and responses from the primate ventral stream were

---

**Algorithm 1** Calculate sparse manifold capacity with Center Correlation

---

**Function:** sparse_capacity_with_correlation $(X)$
**Input data:** Set of manifolds $X = \{M^1, M^2, ..M^P\}$, $M^i \in \mathbb{R}^{Dxm}$
**Output data:** sparse manifold capacity $\alpha_c$

1: $\{\mu_1, \mu_2, ..\mu_P\} \leftarrow$ local_linear_centroids$(X)$
2: Sample $\Delta \subset \{\mu_i - \mu_j | i \neq j, i, j \in \{1, ..P\}\}$
3: Compute common component matrix, $V$, using $\Delta$ ((Cohen et al., 2020) Suppl)
4: **for** manifold $M^i$, $i \in \{1, 2...P\}$ **do**
5:     Compute residual manifold $M_r^i = M^i - V(V^T M^i)$
6:     Compute $D_{gi}, R_{gi}$ of $M_r^i$ (by equation 3)
7: **end for**
8: **for** manifold $M^i$, $i \in \{1, 2...P\}$ **do**
9:     Compute $\alpha_i(\kappa, \vec{D_g}, \vec{R_g})$ (by equation 5)
10: **end for**
11: Compute $\alpha_c = \langle \alpha_i \rangle_i$

---

**Function:** local_linear_centroids $(X)$
**Input data:** Set of manifolds $X = \{M^1, M^2, ..M^P\}$, $M^i \in \mathbb{R}^{Dxm}$
**Output data:** Local linear centroids $\{\mu_1, \mu_2, ..\mu_P\}$, $\mu_i \in \mathbb{R}^{Dx1}$

1: **for** manifold $M^i$, $i \in \{1, 2...P\}$ **do**
2:     Perform LLE on $M^i$ and project $M^i$ to its LLE space.
3:     Let $L^i$ be the projected manifold, and compute the center $\nu_i = \langle L^i \rangle$
4:     Compute $\mu_i$, the projection of $\nu_i$ in the original, higher dimensional space. (See A.2.1)
5: **end for**

---

**Algorithm 2** Nearest Neighbor Algorithm for Computing Local Linear Centroids

---

**Input data:** Manifold $M^i \in \mathbb{R}^{Dxm}$; the projection of $M^i$ after LLE $L^i \in \mathbb{R}^{dxm}$; the mean of $L^i, \nu_i \in \mathbb{R}^{dx1}$
**Output data:** Local linear centroid $\mu_i \in \mathbb{R}^{Dx1}$
**Parameters:** $K \in \mathbb{Z}$

1: Compute the set of $\{\delta_j = \|\nu_i - L_j^i\| | \forall j \in \{1, ...m\}\}$
2: $\delta_{j1}, ...\delta_{jK} \leftarrow \min_{firstK}\{\delta_j\}$
3: $\mu_i = \langle M_j^i \rangle_{j \in \{j1, j2, ...jK\}}$

---

**Algorithm 3** Weighted Average Algorithm for Computing Local Linear Centroids

---

**Input data:** Manifold $M^i \in \mathbb{R}^{Dxm}$; the projection of $M^i$ after LLE $L^i \in \mathbb{R}^{dxm}$; the mean of $L^i, \nu_i \in \mathbb{R}^{dx1}$
**Output data:** Local linear centroid $\mu_i \in \mathbb{R}^{Dx1}$
**Parameters:** $s \in \mathbb{R}^+$

1: $\forall j \in \{1, ...m\}$, compute $\delta_j = \|\nu_i - L_j^i\|$
2: $\forall j \in \{1, ...m\}$, compute $w_j = 1/\delta_j^s$
3: $\forall j \in \{1, ...m\}$, compute $\hat{w}_j = w_j / \sum_k w_k$
4: $\mu_i = \sum_{j=1}^m \hat{w}_j M_j^i$

---

recorded. Again, responses to inputs of the same class form an object manifold. The Majaj dataset contains neural responses to 64 different classes from the V4 and IT regions. The Freeman dataset contains neural responses to 30 different classes from the V1 and V2 regions.

### A.3.1 DNN Training

A regular VGG-16 was trained using pytorch on the Cifar100 dataset. The model was trained on a stochastic gradient descent optimizer with a learning rate of 0.01 and a momentum rate of 0.9.

A learning scheduler was used to reduce learning rate by 0.5 every 10 epochs. The final training accuracy was nearly 0.90.

### A.3.2 Generating smooth and pointcloud manifolds

Smooth manifolds are generated by affine transformations of an image. For a Cifar100 image, a 24x24 frame is translated randomly from the center of the image by a maximum of 4 pixels horizontally and vertically. Each pointcloud manifold consists of the top 50 well-trained examples from that class. For Figure 3, we extract smooth manifolds from the following layers of the VGG-16

```
['layer_00_Input', 'layer_01_Conv2d', 'layer_02_ReLU',
 'layer_03_Conv2d', 'layer_04_ReLU', 'layer_07_ReLU',
 'layer_09_ReLU', 'layer_10_MaxPool2d', 'layer_12_ReLU',
 'layer_13_Conv2d', 'layer_14_ReLU', 'layer_16_ReLU',
 'layer_21_ReLU', 'layer_23_ReLU', 'layer_24_MaxPool2d',
 'layer_26_ReLU', 'layer_27_Conv2d', 'layer_28_ReLU',
 'layer_30_ReLU', 'layer_37_ReLU']
```

We extract our pointcloud manifolds from the following layers of the VGG-16

```
['layer_00_Input', 'layer_02_ReLU', 'layer_04_ReLU',
 'layer_07_ReLU', 'layer_09_ReLU', 'layer_14_ReLU',
 'layer_16_ReLU', 'layer_21_ReLU', 'layer_23_ReLU',
 'layer_28_ReLU', 'layer_30_ReLU', 'layer_37_ReLU']
```

### A.3.3 Neural Data

The data from Majaj et al. (2015a) are made available by Brainscore (Schrimpf et al., 2020) (Schrimpf et al., 2018). The data consists of neural responses from the V4 and IT regions of four Macaque monkeys. The stimulus set consists of 64 classes, and there are a total of 88 neurons recorded in the V4 region and 168 neurons recorded in the IT region. The data from Freeman et al. (2013) are made available by the authors of the work. There are a total of 102 and 103 neurons recorded from the V1 and V2 regions of 13 Macaque monkeys respectively. The set of stimuli contains 15 texture classes. For the purpose of the author's experiments, the 15 texture classes are used to generate corresponding 15 noise classes. We consider the neural data corresponding to texture and noise images together. Hence, in total, we had 30 classes for this dataset.

### A.4 Computing sparse manifold capacity via CLLD vs method in Cohen et al

The calculation for sparse manifold capacity assumes that manifolds are uncorrelated. Cohen et al. (2020) takes into account of correlation for the classical notion of manifold capacity by finding a space of low correlation. This space is computed using the centroids of each manifold. Compared to Cohen et al. (2020), our method outlined by algorithm 1 uses CLLD, categorical local linear differences, to compute the space of low correlation. Figure 7 shows that the method via CLLD is in fact superior to the method in Cohen et al. (2020) for predicting sparse manifold capacity in our neural datasets. Using CLLD allows a better match between theoretical and simulated sparse capacity.

### A.5 Sparse Replica Manifold Analysis Computation Time

Table 2 shows the computational time of Sparse Replica Manifold Analysis for layers of VGG-16 for 40 smooth manifolds. Observe that radius and dimension are computed simultaneously as capacity in the dense label regime. Also, observe that in the regime of sparse label, the compute time of capacity dominates the total compute time.

### A.6 Simulating Capacity

The algorithm for simulating sparse manifold capacity uses a bisection search to find $N_c$, the number of feature dimensions the input data can be reduced to such that half of the one-vs-rest dichotomies are linearly separable. The full algorithm is in Algorithm 4.

Figure 7: This figure shows the match between simulated and theoretical sparse manifold capacity on neural datasets when different methods are used to take into account of manifold correlation. In **(a)**, the method presented in Cohen et al. (2020) is employed. The manifold centroids are used to recover the space of low correlation. In **(b)**, CLLD are used to recover the space of low correlation. Refer to appendix A.2.

Table 2: Sparse Replica Manifold Analysis Computation Time (seconds) for 40 smooth manifolds in VGG-16 Layers

| Vgg16 Layers | **Dense Labels** | | | **Sparse Labels** | | |
|---|---|---|---|---|---|---|
| | $R,D$ | Capacity | Total Time | $R_g,D_g$ | Capacity | Total Time |
| Input | 51.68 | **51.68** | 159.99 | **1.01** | 771.75 | 857.46 |
| ReLU-2 | 50.21 | **50.21** | 111.09 | **1.24** | 762.17 | 851.72 |
| ReLU-4 | 50.00 | **50.00** | 99.41 | **1.16** | 846.37 | 932.27 |
| ReLU-7 | 50.30 | **50.30** | 84.51 | **1.23** | 832.91 | 918.44 |
| ReLU-9 | 50.92 | **50.92** | 82.18 | **1.20** | 768.40 | 856.76 |
| ReLU-14 | 48.53 | **48.53** | 68.31 | **1.14** | 715.54 | 802.85 |
| ReLU-16 | 48.34 | **48.34** | 69.76 | **1.21** | 798.93 | 893.71 |
| ReLU-21 | 46.34 | **46.34** | 69.98 | **1.15** | 1336.58 | 1424.74 |
| ReLU-23 | 46.34 | **46.34** | 61.31 | **1.28** | 1153.42 | 1245.36 |
| ReLU-28 | 45.76 | **45.76** | 62.43 | **0.76** | 1290.43 | 1379.53 |
| ReLU-30 | 45.97 | **45.97** | 81.88 | **0.82** | 1086.01 | 1177.79 |
| ReLU-37 | 45.10 | **45.10** | 105.36 | **1.07** | 1198.15 | 1284.02 |

## A.7 CAPACITY FOR ELLIPICAL MANIFOLDS

Figure 6 measures simulated capacity while varying manifold effective dimension. Effective dimension is equivalent to the manifold Gaussian dimension if the manifold is an ellipsoid (Chung et al., 2018). Hence, simulated capacity for figure 6(c,d) is calculated assuming manifolds have an elliptical structure. As such, we use the Maximum Margin Manifold Machines algorithm described in (Chung, 2017).

## A.8 MEAN FIELD THEORY OF MANIFOLD SEPARATION

In this section, we elucidate the emergence of anchor points in the solution for manifold capacity. Following Gardner's framework, Chung et al. (2018) determined that in the max margin solution for a homogeneous system of manifolds, inverse manifold capacity is given by $\alpha_M^{-1} = \langle F(\vec{T}) \rangle_{\vec{T}}$, where $F(\vec{T}) = \min_{\vec{V}} \{ \|\vec{V} - \vec{T}\|^2 | g_S(\vec{V}) - \kappa \geq 0, \forall \vec{S} \in S \}$ and $g_S(\vec{V}) = \min_{\vec{S}} \{ \vec{V} \cdot \vec{S} | \vec{S} \in S \}$.

$\alpha_M$ is interpreted for a single manifold. $\vec{T}$ are sampled Gaussian vectors while $\vec{V}_i = y\vec{w} \cdot \vec{u}_i$ ($i = 1, ...D + 1$), where $\vec{w}$ is a solution hyperplane, $\vec{u}_i$ is the manifold's ith basis vector, and $y$ is the

---

**Algorithm 4** Simulated Capacity

---

**Function:** bisection_search $(X)$
**Input data:** Set of manifolds $X = \{M^1, M^2, ..M^P\}$, $M^i \in \mathbb{R}^{Dxm}$
**Output data:** sparse manifold capacity $\alpha_c$

1: Initialize $N = 0$ and $Found = False$
2: Initialize $N_{max} = D$ and $N_{min} = 2$
3: $N_{next} \leftarrow (N_{max} + N_{min})/2$
4: $prob_{max} \leftarrow$ is_linearly_separable$(X, N_{max})$
5: $prob_{min} \leftarrow$ is_linearly_separable$(X, N_{min})$
6:
7: **while** not Found and $N! = N_{next}$ **do**
8:     $N \leftarrow N_{next}$
9:     $prob \leftarrow$ is_linearly_separable$(X, N)$
10:
11:     **if** $\|prob - 0.5\| < tol$ **then**
12:         Found$\leftarrow$ True
13:
14:     **end if**
15:
16:     **if** $(prob_{min} - 0.5) * (prob - 0.5) < 0$ **then**
17:         $N_{max} \leftarrow N$
18:         $prob_{max} \leftarrow prob$
19:
20:     **else**
21:         $N_{min} \leftarrow N$
22:         $prob_{min} \leftarrow prob$
23:
24:     **end if**
25:     $N_{next} \leftarrow (N_{max} + N_{min})/2$
26:
27: **end while**
28:
29: **if** Found **then**
30:     $N_c \leftarrow N_{next}$
31:
32: **else**
33:     Interpolate for $N_c$
34:
35: **end if**
36: $\alpha_c = P/N_c$

---

manifold's label. $\vec{V}$ is described as the signed fields induced by the solution hyperplane, and $S$ can be thought of as the set of points on the manifold. The KKT interpretation is to find $\vec{V}$ given by

$$\vec{V} = \vec{T} + \lambda \tilde{S}(\vec{T}) \tag{6}$$

such that the following holds

$$\lambda \geq 0$$

$$g_S(\vec{V}) - \kappa \geq 0$$

$$\lambda[g_S(\vec{V}) - \kappa] = 0$$

and $\tilde{S} = \arg\min_{\vec{S} \in S} \vec{V} \cdot \vec{S}$. The mean field interpretation of equation 6 is that $\vec{V}$ is a decomposition of contributions from other manifolds represented by $\vec{T}$ and the fixed manifold represented by $\tilde{S}$. Furthermore, equation 6 implies that the resulting $F(\vec{T})$ is precisely $\|\lambda \tilde{S}(\vec{T})\|^2$, with $\tilde{S}(\vec{T})$ denoted as the anchor point of the manifold.

---

**Function:** is_linearly_separable $(X,N)$
**Input data:** Set of manifolds $X = \{M^1, M^2, ..M^P\}$, $M^i \in \mathbb{R}^{Dxm}$; Number of feature dimensions $N$
**Output data:** Probability of separation given dichotomy, $prob$

1: Initialize $sep = 0$
2: Project $D$-dimensional manifolds to $N$ dimensions using random projection.
3:
4: **for** $i \in \{1, ..., P\}$ **do**
5:     Assign the set $Y$ of binary labels for each manifold, where manifold $i$ gets the positive label.
6:     Find consistent SVM solution for input $X$ and label $Y$
7:
8:     **if** solution exists **then**
9:         $sep = sep + 1$
10:
11:     **end if**
12:
13: **end for**
14: $prob \leftarrow sep/P$

---

### A.9 HOMOGENEOUS VS HETEROGENEOUS GEOMETRIES

We've made the claim in the previous section, Derivation to Theorem 1, that the equation 4 is insufficient for a system of manifolds with varying radii and dimensions. We demonstrate the disparity in the match between the theoretical and simulated sparse manifold capacity when different assumptions of the geometries are made in figure 8 and 9. Using the smooth manifold and the pointcloud manifold data from the VGG-16 dataset used throughout this paper, we compute the capacity using homogeneous geometries by averaging across the Gaussian radii and dimensions of all manifolds in the system. Comparing the left and the right plots in each picture, it is evident that assuming homogeneous geometries is insufficient in a system where manifolds have varying Gaussian radii and dimensions.

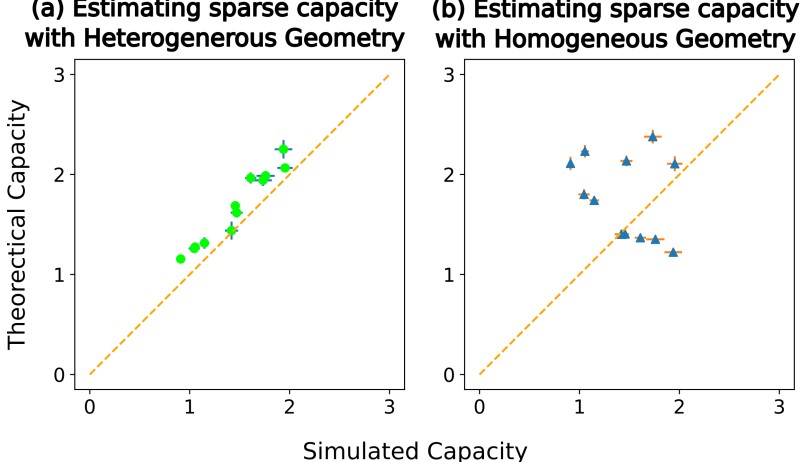

Figure 8: **Heterogeneous vs Homogeneous geometries in estimating sparse manifold capacity for smooth manifolds.** (a) is the result of incorporating the heterogeneous geometries in sparse manifold capacity as demonstrated in Fig 3a of the paper (i.e. using theorem 1). (b) is the result of assuming homogeneous geometries to estimate sparse manifold capacity (i.e. equation 4 in the paper). If we simply use the average dimension and radius of all manifolds to estimate the sparse capacity, we have the resulting (mis)match in (b)

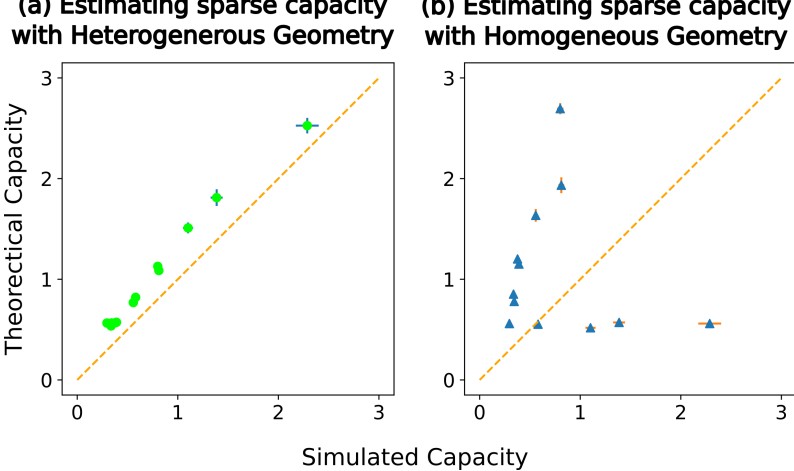

Figure 9: **Heterogeneous vs Homogeneous geometries in estimating sparse manifold capacity for pointcloud manifolds.** (a) is the result of incorporating the heterogeneous geometries in sparse manifold capacity as demonstrated in Fig 3b of the paper (i.e. using theorem 1). (b) is the result of assuming homogeneous geometries to estimate sparse manifold capacity (i.e. equation 4 in the paper). If we simply use the average dimension and radius of all manifolds to estimate the sparse capacity, we have the resulting (mis)match in (b)

### A.10 Sparse Replica Manifold Anaysis on Imagenet-trained Resnet 101

We used an imagenet-trained Resnet101 and create 20 object smooth manifolds from the following extracted convolutional layers. Results of the match between theoretical and simulated sparse manifold capacity is shown in figure 10

```
[ 'layer_123_Conv2d', 'layer_014_Conv2d', 'layer_155_Conv2d',
  'layer_181_Conv2d', 'layer_202_Conv2d', 'layer_232_Conv2d',
  'layer_028_Conv2d', 'layer_048_Conv2d' ]
```

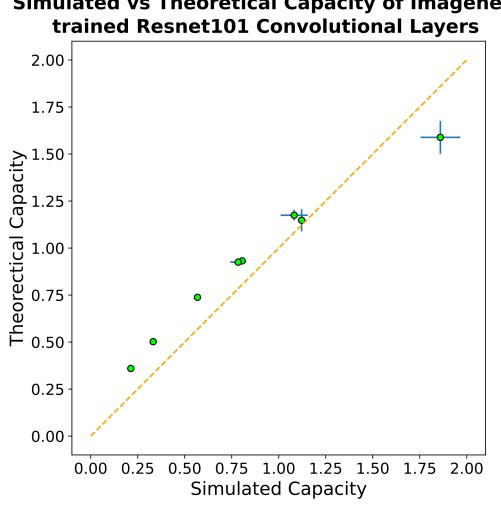

Figure 10: Theoretical vs simulated sparse manifold capacity measured on 20 manifolds from an imagenet-trained Resnet101.

## A.11 CLLD SENSITIVITY TO NUMBER OF NEURONS

Estimating sparse manifold capacity requires accounting for manifold correlation. Cohen et al. (2020) successfully account for manifold center correlation in their work that applied dense labeled manifold capacity to artificial neural network. In appendix A.4 and figure 7 using the method in Cohen et al. (2020) is insufficient for sparse manifold capacity. Instead, we show that the method we presented in this paper using CLLD does a better job. In figure 11, we show that using CLLD always outperform Cohen et al regardless of the number of neurons present in the data. We vary the number of neurons by random projection. The results show that the estimation for sparse manifold capacity is robust.

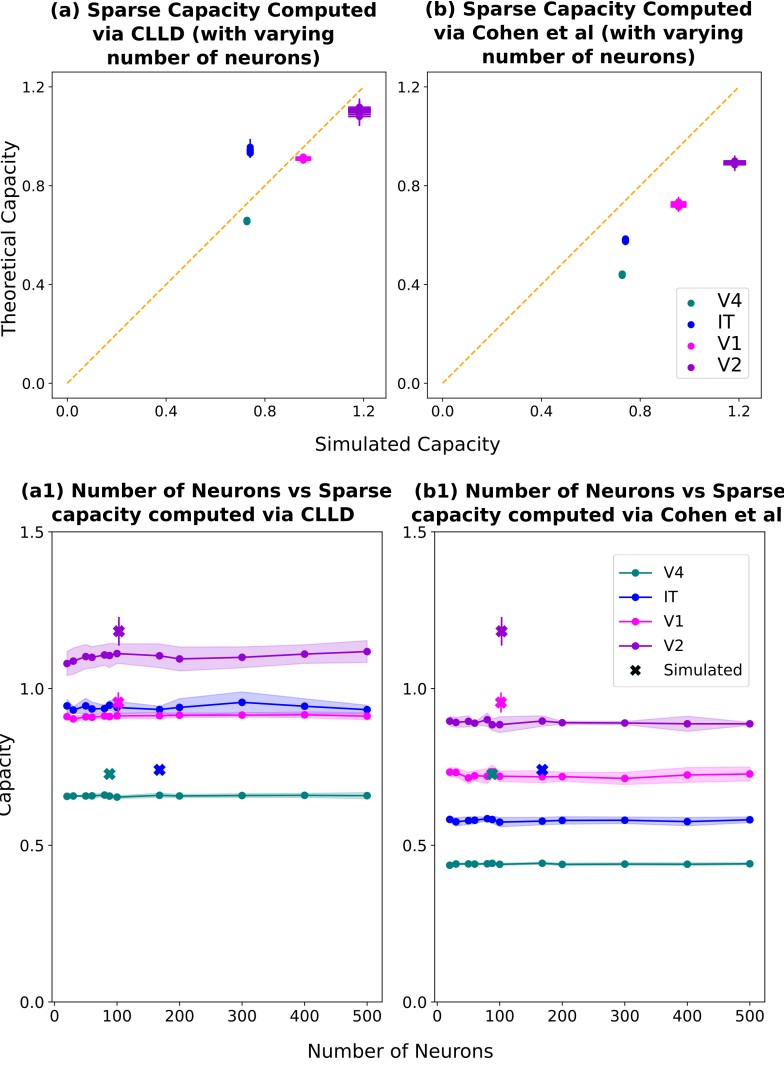

Figure 11: In (a) and (b), each color represents the capacity at various layers of the ventral stream. Each point of the same color is the capacity measured from randomly projecting the neural manifold to some number of features. (a1) and (b1) show explicitly the number of neurons vs the resulting estimated capacity. We vary the number of neurons by random projection. The resulting theoretical capacity changes little as a result. Furthermore, compared to the method in Cohen et al (b1), the method via CLLD is closer to the ground truth capacity (represented by x).

