# OpenReview forum: "Efficient neural representation in the cognitive neuroscience domain: Manifold Capacity in One-vs-rest Recognition Limit"
_ICLR.cc/2023/Conference — Submitted to ICLR 2023_

### Official Review · Reviewer_4A5K · 2022-10-27

**Confidence:** 3
**Correctness:** 4
**Technical Novelty And Significance:** 3
**Empirical Novelty And Significance:** 3
**Recommendation:** 6

**Clarity, Quality, Novelty And Reproducibility:**

Nits:

The abstract could be a bit more high level, without references or footnotes and more accessible for a wider audience.

Throughout the paper references seem split, are they not in the same \cite{} call in latex?

Footnote 1: point missing.

Introduction, 3 paragraph, last sentence: this is unclear, please explain/rephrase

Define VC dimension when first using the term (also Cover)

Intro, last paragraph before list: Does that mean manifold capacity

Section 2: Please define \tilde{s} this is important!

Sec 3: ‘We extend this work by’… -‘of’ correlation+’s’

3.1. First sentence manifold+’s’

‘Furthermore, this bias’ …. Why? Please clarify

3.1.2: what does ‘null space of low correlations’ mean? Clarify

Fig 2. How do you project back?

4.1. What is theoretical vs simulated capacity and why is one of them ground truth??

Fig 3c: please change colors, these are too similar

Fig 4: really cool!

4.5 first paragraph: this is a nice intuition!

Fig 5 column iv: this seems add odds between ANN and biological data. Is that significant, and if yes, can you speculate why?

Fig 6a: Great plot, but please improve the clarity of the caption



**Strength And Weaknesses:**

Strength: The paper is clearly written and presents and interesting extension to prior work that is well matched to realistic settings. The experiments are straightforward and support the claims.

Weaknesses: It would be great if the methods would rely less on prior work. Basically, it is impossible to properly understood the current paper fully without having read the prior work by Chung et al. This is unfortunate and might make the current work seem more incremental.

**Summary Of The Paper:**

I thank the authors for their thoughtful replies as well as the clarifications that they have made in the paper. I would increase my score to 7 if that were an option. In any case, I think that this is a valuable contribution and look forward to seeing the paper published =)

This paper extends prior work on measuring manifold capacity to the setting of sparse one-vs-all classification. This is well matched to biological data and provides a more optimistic bound on the number of neurons needed to compute the proposed metrics. They show that across artificial and biological processing stages manifold dimension decreases while ambient space size might increase – both factors that contribute to superior classification performance.

**Summary Of The Review:**

Overall, I think that this is a cool extension for measuring manifold capacity in sparse settings. The results across processing hierarchies also make sense in artificial and biological systems (maybe a newer architecture like ResNet would have been more interesting than VGG). It would be great if the paper were more self-contained, currently it seems somewhat derivative/incremental on prior work. That being said, I think improving previous bounds to realistic sparse settings is really cool and valuable.

---

> ### Author Response · Authors · 2022-11-17
> **Response part 1**
>
> We are glad the reviewer finds our work “interesting” and “well matched to realistic settings.” We further appreciate the suggestions made by the reviewer to make our paper more clear and happily incorporated the changes necessary to our figures, captions, and text. We clarified definitions in section 2 on the theory and we included more detailed information in our appendix. We address the specific changes we made:
>
> **Introduction, 3 paragraph, last sentence: this is unclear, please explain/rephrase**
>
> We’ve rephrased the sentence from “This implies a bias for lower manifold intrinsic dimension to achieve invariance and robustness. “ to
> > “This observation suggests feature representations with a low intrinsic dimension allows invariance and robustness in classification.”
>
> We hope the edit has a clearer meaning. This sentence follows the results shown in Cohen et al, where capacity for ANN was found to increase across layers of the ANN as dimension and radius decrease. Cohen et al concluded that manifolds with low intrinsic dimension are correlated with better separability and thus classification.
>
> **Define VC dimension when first using the term (also Cover)**
>
> We have included the definition of VC dimension and Cover’s theorem. We have changed the sentence “VC dimension  and Cover's theorem study the worst and average cases of separability and consider all random dichotomies” to
> > “VC dimension in learning theory returns the largest number of points in a set such that every dichotomy of the set can be learned by a given hypothesis function…. Cover's theorem returns the number of linearly separable sets given $N$ points and in a $D$ dimensional space...”
>
> **Section 2: Please define \tilde{s} this is important!**
>
> Thank you for bringing our attention to this important point. $\tilde{s}$, called the anchor point, is a point on a manifold or its convex hull that contributes to the linear hyperplane. It emerges from solving the mean field equations for the capacity of linear manifold separation explained in Chung et al and is the solution to the KKT interpretation. \tilde{s} of a given manifold is determined uniquely by the orientation and location of other manifolds, represented by the Gaussian vector $T=(t,t_0)$. Hence, we sample Gaussian vectors to gather statistics of the resulting anchor points and gauge the geometries of the manifold. We clarified around this point in section 2 of the updated version of the paper and included a section in the appendix A.8. on interpreting the anchor points as the solution to the KKT interpretation (see this section here as well: https://imgur.com/kIbBjTr)
>
> > “To determine $\alpha_M$... involves finding manifold anchor points,
> $\tilde{s}(\vec{t},t_0)$, uniquely determined by Gaussian vectors $T\in \mathbb{R}^{D+1}$, … Given some dichotomy and orientation of the manifolds, the anchor point is a point on the manifold or its convex hull that contributes to the separating linear hyperplane. The notion of anchor point emerges from the solution to the KKT interpretation of the mean field equations for capacity, discussed in depth in \cite{chung2018classification} and described in the appendix A.8. Similar to support vectors in a Support Vector Machine, these anchor points change on the location and labels of other manifolds. The Gaussian vectors represent the variability in orientations, locations, and labels of other manifolds.”
>
> **“‘Furthermore, this bias’ …. Why? Please clarify”**
>
> Under theorem 1, calculating the sparse manifold capacity requires maximizing capacity through a bias parameter. The bias parameter adds to the margin for the positive labeled manifold and subtracts from the margin for the population of negative labeled manifolds. The resulting effect on the margin, described in depth in Chung et al, puts the manifolds in a regime such that for each manifold, t_0 becomes negligible to the anchor point, i.e. the relevant manifold geometries become the Gaussian radius and dimension. We clarified this in section 3.1 of the revised version. In the main text, in placed of the statement in question, we wrote:
> > “The resulting effect of the bias parameter, shown in \cite{chung2018classification}, puts
> the manifolds in regime where $t_0$ becomes negligible to the anchor point of a
> contributing manifold.”
>
> **“what does ‘null space of low correlations’ mean? Clarify”**
>
> We thank the reviewer for pointing this mistake out. As we have mentioned to another reviewer, this was in fact a typo in the draft. The correction is “space of low correlations” in the main text. The phrase “null space” is meant towards the null space of a set of orthonormal vectors, called the common components, such that the manifold centers projected into this null space will have a diagonal correlation structure. Cohen et al provided the algorithms to recover the common components in their work. Projecting the entire manifold into the null space recovers the manifold with almost no correlations.

---

> > ### Author Response · Authors · 2022-11-17
> > **Response part 2**
> >
> > **“Fig 2. How do you project back?”**
> >
> > The methods used to project a point from a low-dimensional space to the high-dimensional space are discussed in appendix A.2.1. We described the procedure used for the experiments in our paper in algorithm 2 (shown https://imgur.com/eP0chxz). Given the center of the manifold in the low-d space, we compute its distance to the other points of the manifolds in the same low-d space. Then taking the indices of the K closest points, we approximate the local linear center in the high-d space by taking the average of those corresponding K points in the high-d space. In this algorithm, our manifold has dimension Dxm, where D is the dimension of the ambient feature space and m is the number of points that make up our manifold (i.e. the number of examples in the class that the manifold represents). The examples in the manifold is indexed by j.
> >
> > **“What is theoretical vs simulated capacity and why is one of them ground truth?”**
> >
> > The theoretical capacity is the sparse manifold capacity given by theorem 1. Simulated capacity is the empirical capacity of the manifolds found via a bisection section for the critical number of features over which the probability of linearly separating a one-vs-rest dichotomy falls from 1 to 0. In practice, we search for the number of features where the probability is 0.5. Hence, we say the simulated capacity is the ground truth as we try to estimate it using theorem 1. We clarify around this point in section 4.1:
> > > “In this section, we show how well theorem 1 predicts the sparse manifold capacity. Figure ...
> > demonstrates the match between the theoretical sparse manifold capacity (determined by our theorem)
> > and simulated sparse manifold capacity (the ground truth determined empirically) for responses in
> > layers of a CIFAR-100 trained VGG-16 and brain regions of Macaque monkeys to given stimuli.”
> >
> > **“Fig 5 column iv: this seems add odds between ANN and biological data. Is that significant, and if yes, can you speculate why?”**
> >
> > We appreciate the reviewer pointing this significance out. To our knowledge, this paper is the first time where manifold capacity is validated on neural data and we are able to observe the manifold geometry in neural data. The differences between the ANN and the ventral stream could elucidate differences between ANN models and real neural data, which can help guide us to better models of the visual cortex. This emphasizes the use of sparse manifold capacity and its geometries as a metric of similarity between ANN and the brain. We have highlighted this point in section 4.4 where the figure is presented.
> > > “...manifold Gaussian dimension and effective dimension appear to exhibit the opposite
> > behavior relative to the artificial network. This suggests difference between the artificial
> > and the biological brain and points toward using manifold geometries as a metric of finding and fitting an artificial brain model“
> >
> > **Throughout the paper references seem split, are they not in the same \cite{} call in latex?**
> >
> > Thank you for pointing this out. We have fixed this in the updated paper.

---

### Official Review · Reviewer_TxMd · 2022-10-27

**Confidence:** 4
**Correctness:** 3
**Technical Novelty And Significance:** 4
**Empirical Novelty And Significance:** 4
**Recommendation:** 8

**Clarity, Quality, Novelty And Reproducibility:**

The paper is well-written and organized. Empirical discoveries in this paper are highly novel and would be beneficial to both computational neuroscience and machine learning community. Although implementation details of the proposed approach are described in the Appendix, authors could anonymously release the code for increased reproducibility.

**Strength And Weaknesses:**

Strengths
- The paper is overall very complete in itself, especially the inclusion of MFTMA in the main manuscript is very helpful.
- Thorough empirical study is performed.
- Nice to see that sparse analyses are significantly faster compared o dense analyses.
- Overall clarity and organization of the paper are great.


Weaknesses
- 2.0  (t^tilda, t_0) vectors are lacking variable definition. Also, why are these sampled from a Gaussian distribution?
- Appendix section A.1 DERIVATION TO THEOREM 1 may be included in the main manuscript to improve completeness. I regard this derivation as one of the main technical novelties of this paper.
- I understand that Eqs 4 and 5 are built upon the assumption that the manifolds are distributed Gaussian with a given radius and Dimension. What happens when they are not? Does this analysis break down?
- 3.1.2 "null space of low correlations" what does that mean? Please give some background.
- 4.1, "Observe this means that a manifold needs to have enough feature dimension for the bisection search to operate appropriately." It is not only a hard sentence to understand, but why this should be the case for a 0.5 probability that a one-vs-rest dichotomy is separable is unclear. For completeness, the authors could elaborate on this.
- Figure 3 indicates that manifold capacity is underestimated on artificial data, whereas it is almost exactly estimated in real neural data. A discussion on why this might be the case could be interesting.
- It would be interesting to see the analysis on some other (possibly larger) datasets, such as ImageNet. Also, analysis with a more recent and widely used deep architecture would be more relevant, such as Residual CNNs and/or transformers.


**Summary Of The Paper:**

The authors of the paper propose Sparse Replica Manifold analysis for estimating sparse manifold capacity that measures how many object manifolds can be separated under sparse classification tasks, which is pervasively used in both neuroscience (real neural data) and deep learning (artificial neural data) domains. Building upon Mean Field Theory Manifold Analysis (MFTMA), the authors derive and prove Theorem 1, which theoretically computes sparse manifold capacity under the assumption of different Gaussian Radii and Dimensions for each manifold. To account for the correlation between manifolds, authors leveraged CLLD to compute a space of low correlation. The authors then perform comprehensive experiments on both real and artificial neural data to demonstrate that their theoretical and empirical analyses are consistent. Overall, the paper is presented very well and has convincing results.

**Summary Of The Review:**

This study on neural representations is very intriguing for both discriminatory analyses in statistical learning and to better study computational principles in neuroscience. Therefore, studies like this are (and will be) of interest to venues like ICLR. I suggest acceptance of this work. I believe that if the authors address/comment on the concerns mentioned in the Weaknesses section, the overall strength of the paper will be increased.

---

> ### Author Response · Authors · 2022-11-17
> **Response part 1**
>
> We are glad the reviewer finds our work “highly novel” and “beneficial to both computational neuroscience and the machine learning community.” We appreciate the invaluable suggestions and incorporated changes in our paper to make the writing clear.
>
> **“(t^tilda, t_0) vectors are lacking variable definition. ”**
>
> We have clarified the definition of (t,t_0) in the main text, section 2:
> > “....Gaussian vectors T in R(D+1), whose components T_i are sampled from N (0, 1). For the rest of this paper, we denote T = (t, t0).”
>
> **“Also, why are these sampled from a Gaussian distribution?”**
>
> Given a fixed manifold, its anchor point is determined by the orientations,  location, and the label of the other manifolds. The Gaussian vectors sampled represent the variability in the orientation of other manifolds around a fixed manifold and uniquely determine the anchor point on the fixed manifold given variability attributed by the other manifolds. We clarified around this point in section 2 of the revised version of the paper, including the lines:
> > “Given some dichotomy and orientation of the manifolds, the anchor point is a point on
> the manifold or its convex hull that contributes to the separating linear
> hyperplane…Similar to support vectors in a Support Vector Machine, these anchor
> points change based on the location and labels of other manifolds. The Gaussian
> vectors represent the variability in orientations, locations, and labels of other manifolds“
>
> **“Appendix section A.1 DERIVATION TO THEOREM 1 may be included in the main manuscript to improve completeness. I regard this derivation as one of the main technical novelties of this paper.”**
>
> We are grateful for this suggestion and included a shortened version of the derivation in section 3.1.2 of the revised version of the paper. The entire derivation remains in appendix A.1 for detailed reference.
>
> **“I understand that Eqs 4 and 5 are built upon the assumption that the manifolds are distributed Gaussian with a given radius and Dimension. What happens when they are not? Does this analysis break down?”**
>
> Yes, the equations mentioned assume that the manifolds are distributed Gaussian, i.e. randomly located and oriented. In realistic data, the equations would be insufficient due to correlations in real data. Prior work by Cohen et al was able to account for manifold center correlations in their analysis of manifold capacity in artificial neural networks, but their method is insufficient for sparse manifold capacity. Shown in appendix A.4, the figure (also shown here https://imgur.com/GhqqdT4) shows that the method used by Cohen et al (a) is less robust towards matching the theoretical and simulated sparse manifold capacity. Instead, the method we described in the paper using CLLD allows a better match, however, there can be other correlation terms not accounted for. Full treatment of further corrections is outside the scope of this paper, and is an important direction for future study. We hope the results of this work can be utilized in such future study.
>
> **"null space of low correlations" what does that mean? Please give some background.”**
>
> We thank the reviewer for pointing this mistake out. This was in fact a typo in the draft. The correction is “space of low correlations” in the main text. The phrase “null space” is meant towards the null space of a set of orthonormal vectors, called the common components, such that the manifold centers projected into this null space will have a diagonal correlation structure. Cohen et al provided the algorithms to recover the common components in their work. Projecting the entire manifold into the null space recovers the manifold with almost no correlations.

---

> > ### Author Response · Authors · 2022-11-17
> > **Response part 2**
> >
> > **"Observe this means that a manifold needs to have enough feature dimension for the bisection search to operate appropriately." It is not only a hard sentence to understand, but why this should be the case for a 0.5 probability that a one-vs-rest dichotomy is separable is unclear. For completeness, the authors could elaborate on this.**
> >
> > We thank the reviewer for pointing this out. What we meant to suggest is that since we use a bisection search to find the critical number of features, it is necessary that the manifold lives in a high enough dimensional space. In particular, when we determine simulated capacity, we use random projection to reduce the number of features and use a hard margin SVM to determine if the manifolds are separable on some given dichotomy. Capacity is determined as P/N_c (P is the number of manifolds and N_c is the critical number of features (over which the probability a random dichotomy is separable turns from 1 to 0.). The critical number of features N_c is interpolated between the number of features where almost all the dichotomies are separable and where none of them are. Hence, it is necessary that we have at least one projected dimension where the manifolds are mostly linearly separable. If in the full feature space, the manifolds are inseparable, we are unable to interpolate. We clarified around this in the updated section 4.1. We replaced the sentence in question and added:
> > > “The ground truth sparse manifold capacity is determined by interpolating the critical
> > number of features (Nc) over which the probability a one-vs-rest dichotomy is linearly
> > separable drops from 1 to 0. In practice, we use a bisection search for Nc where the
> > probability is nearly 0.5.“
> >
> > **“Figure 3 indicates that manifold capacity is underestimated on artificial data, whereas it is almost exactly estimated in real neural data. A discussion on why this might be the case could be interesting.”**
> >
> > We acknowledge that sparse manifold capacity appears to be better estimated on the neural data, however, there are some layers in the artificial neural network where capacity is well estimated as shown in Figure 3a and 3b. A speculation on why the neural data behaves better could be that neural data have some underlying neural noise that smooths out the neural manifold geometries, making it more amenable to the analysis. Meanwhile, the artificial data is deterministic and driven by stimulus variability. In the future, we can explore analyzing manifold capacity on stochastic neural networks. We have added these notions to section 4.1

---

> > > ### Comment · Reviewer_TxMd · 2022-11-29
> > > **Response to authors**
> > >
> > > I thank the authors for clarifying the relevant points. They enhanced the paper according to the comments and suggestions. I believe that this paper has good quality and merit for publication. I would like to keep my evaluation as "8: accept, good paper".

---

### Official Review · Reviewer_EQzE · 2022-10-30

**Confidence:** 3
**Correctness:** 3
**Technical Novelty And Significance:** 2
**Empirical Novelty And Significance:** 2
**Recommendation:** 3

**Clarity, Quality, Novelty And Reproducibility:**

The paper is relatively well motivated and mostly clear; however, the results appear somewhat incremental and the improvements in their analysis have not been thoroughly dissected.

**Strength And Weaknesses:**


Strengths:

- Moving to case where different manifolds can have different radii and varying correlations is better match to real data.

- Simple algorithms and some theoretical justifications.

- Analysis on two neural datasets show that their approach can better predict properties of neural responses than the homogeneous analysis.

Weaknesses:

- The main contribution of the work is extending previous theory to the heterogeneous manifold case, and providing a new algorithm for computing inter-manifold correlations. Thus, it would be useful to have some ablations to understand how adding different assumptions or constraints impacts the theory and match to practice.

In particular,
(i) How much does the relaxation of the homogeneity constraint on the manifolds impact the match to data vs. the sparsity assumption?
(ii) How does ovr match the assumption about how the brain is performing object recognition? It wasn’t entirely clear how the ovr is important -- or if this analysis tells us something about how neural circuits are trying to classify different objects.
(iii) How much is the low rank projection step in CLLD impacting the match to data? How do you select the dimension to project to?

- Further synthetic experiments involving homogeneity and heterogeneity would be useful to understand the theory

- The extension of the theory is relatively incremental and hasn’t been dissected in a rigorous manner.

- Instead of measuring across manifold correlations using manifold centers, they propose to use categorical local linear differences (CLLD). Are these metrics sensitive to the size of the dataset and how many neurons are in each?


**Summary Of The Paper:**

This paper extends the sparse replica manifold analysis approach in Chung 2018 to incorporate across-manifold correlations and account for non-homogeneous manifolds. They demonstrate the application of sparse manifold capacity allows analysis of a wider class of neural data and show that their theory and algorithms provide a better match to real data.



**Summary Of The Review:**

This paper provides new tools for analyzing the geometry of representations in artificial and biological neural networks. They extend theory from Chung 2018 to a more realistic setting where manifolds can have different underlying geometry and propose an algorithm for computing inter-manifold correlations.

While their analysis and theory does indeed provide better matches to real data, the theoretical results appear to be relatively incremental and haven't been adequately dissected in the empirical studies. Is the better match due to using heterogeneous manifold estimates, or is the proposed estimation method key? The role of the dimension of the projection, the dimensionality of the manifold versus the number of neurons or ambient dimension hasn't been clearly dissected in the experiments.

---

> ### Author Response · Authors · 2022-11-17
> **Response part 1**
>
> We appreciate the helpful feedback from the reviewer and will happily incorporate the suggestions they made in the revised paper. We would like to address their specific questions below:
>
> **“(i) How much does the relaxation of the homogeneity constraint on the manifolds impact the match to data vs. the sparsity assumption?”**
>
> Our work focuses on capacity under the sparse labeling case that was little considered in previous work. Though Chung et al discussed the sparse label case, they tested their theory on artificial data where manifolds share the same geometries and have no center correlations. Thus, the original theory is inapplicable to real data due its constraints. Cohen et al took the first step to apply the theory to real data by accounting for manifold correlations, but only under the dense labeling regime. Our theory fills the gap to incorporate heterogeneous geometries for estimating sparse manifold capacity in real data where manifolds will have varying shapes and geometries.
>
> In appendix A.9, we demonstrate the disparity in the match between the theoretical and simulated sparse manifold capacity when different assumptions of the geometries are made. See plot here: https://imgur.com/IY5BtaT. Using the smooth manifold data from layers of Vgg16, (a) is the result of incorporating the heterogeneous geometries in sparse manifold capacity as demonstrated in Fig 3a of the paper (i.e. using theorem 1). (b) is the result of assuming homogeneous geometries to estimate sparse manifold capacity (i.e. eq 4 in the paper). For (b), we use the average dimension and radius of all manifolds and use them as inputs to estimate the sparse capacity via equation 4, and we have the resulting (mis)match in (b).
>
> More precisely, (dense) manifold capacity is calculated as an average of P alpha_m where P is the number of manifolds and alpha_m can be approximated by equation 2 (https://imgur.com/TEXNcgU) for each of P manifolds. In the sparse case and with homogeneous geometries, we use equation 4 (https://imgur.com/8wCMd8h), which precisely averages across alpha_m, each with inputs from a manifold with either positive labeling or negative labeling indicated by +b and −b. Since all R and D are the same in a homogeneous system, equation 4 sufficiently accounts for all possible assignments of positive and negative labeling. However, when each manifold has a different geometry, equation 4 will be inapplicable and inaccurate as demonstrated by (b).
>
> The derivation to theorem 1 (appendix A.1) illustrates the idea described in the previous paragraph to the insufficiency of equation 4 for the case of heterogenerous geometries. A shorter version is included in the main text (section 3.1.2). We have included the graphics in the appendix A.9 to elucidate this point. We hope this will make our theory and the goals of the paper clear.
>
> **“(ii) How does ovr match the assumption about how the brain is performing object recognition? It wasn’t entirely clear how the ovr is important -- or if this analysis tells us something about how neural circuits are trying to classify different objects.”**
>
> While we do not claim that the brain performs object recognition in a one-vs-rest manner, our work provides the technique to estimate capacity in a relevant setting (one vs rest) for most neuroscience tasks. Previous works, such as Chung et al and Cohen et al, while setting up the stage for calculating capacity, do not address the one vs rest setting that is most relevant in the neuroscience domain and to enable a comparison between neural networks and the human visual cortex. Most common tasks in neuroscience experiments require their subjects perform one-vs-rest classification such as the oddity task (where subjects find the odd object in a group) and delayed matching (where subjects pick the object of a similar class to one they were previously shown). In addition, artificial neural networks use one-hot encodings as label classes and whose accuracy is evaluated on identifying one correct class. Sparse manifold capacity operates under the ovr regime, allowing a direct comparison between brain and ANN and to examine the dynamics in neural data. We have added the following revision to emphasize the relevance of ovr in the fourth paragraph of the introduction:
> > “The regime of object recognition for a classification model and for a monkey performing a delayed matching or oddity task, however, is equivalent to the separation of manifolds on a one-vs-rest basis”
>
> and in paragraph 5 of the introduction:
>
> > “Thus, the sparse labeling regime allows us to apply the theory of manifold capacity to real neural data and use capacity as a measure of recognition and similarity between DNN and the biological brain.“

---

> > ### Author Response · Authors · 2022-11-17
> > **Response part 2**
> >
> > **“(iii) How much is the low rank projection step in CLLD impacting the match to data? How do you select the dimension to project to?”**
> >
> > We thank the reviewer for bringing our attention to this important point. Regarding finding the low dimensional space for the LLE projection step in the algorithm, the experiments in this paper chose the default of 2 components to project to. The heuristic choice was made to minimize the reconstruction error of local linear embedding and maintain the manifold structure. In the case of the artificial neural network, small numbers of components were found to minimize the reconstruction error of the manifolds. We included this detail in the updated version of the paper for reproducibility purposes. In appendix A.2, we wrote
> > > “For the experiments in this paper, the number of components chosen for local linear embedding of each manifold was heuristically chosen as the default value of 2 to minimize the reconstruction error of the dimensional reduction.“
> >
> > We also made a similar footnote in the same text in section 3.1.3
> >
> > We can observe that for the theoretical capacity, minimizing the reconstruction error does not guarantee a match to the simulated capacity, and there could be other sources of correlation. However, despite the heuristic choice, our paper finds that through LLE and CLLD, we can better account for the manifold centroid correlation in the calculations for sparse manifold capacity. The figure shown here (https://imgur.com/GhqqdT4), which is also included in appendix A.4 shows that the method used by Cohen et al (a) is less robust towards matching the theoretical and simulated sparse manifold capacity with correlation. Hence, we can better estimate sparse manifold capacity with manifold correlation via the method with CLLD.
> >
> > **“Instead of measuring across manifold correlations using manifold centers, they propose to use categorical local linear differences (CLLD). Are these metrics sensitive to the size of the dataset and how many neurons are in each?”**
> >
> > We have included an analysis on layers of an imagenet-trained resnet101 in the updated version of the paper (see appendix A.10 or https://imgur.com/gahsWOA) and are working on having more results from applying our sparse replica manifold analysis on larger datasets with varying number of manifolds. (UPDATED: see https://imgur.com/WSoX1j2)
> >
> > For the number of neurons, we have found that using CLLD performs consistently better than using the manifold centers (Cohen et al) to estimate sparse manifold capacity. In this plot (https://imgur.com/oJEKmXM), which is also in appendix A.11 of the revised paper, we show that CLLD allows a better match between simulated and theoretical capacity for several subsets of neurons. In (a) and (b), each color represents the capacity at various layers of the ventral stream. Each point of the same color is the capacity measured from randomly projecting the neural manifold to some number of features. (a1) and (b1) show  explicitly the number of neurons vs the resulting estimated capacity. We vary the number of neurons by random projection. The resulting theoretical capacity changes little as a result. Furthermore, compared to the method in Cohen et al (b1), the method via CLLD is closer to the ground truth capacity (represented by x).

---

### Official Review · Reviewer_yhnc · 2022-11-02

**Confidence:** 2
**Correctness:** 3
**Technical Novelty And Significance:** 3
**Empirical Novelty And Significance:** 3
**Recommendation:** 6

**Clarity, Quality, Novelty And Reproducibility:**

**Clarity**. The paper was not an easy read, as it assumes a lot of prior knowledge in this subdomain.

**Novelty**. The extension appears novel.

** Reproducibility**.No explicit attempts at reproducibility are made in the paper, but a promise to release the method is made.

**Strength And Weaknesses:**

##### Strengths:

1. The idea seems valid, although I do not have the domain knowledge to place its significance.

##### Weaknesses:

1. The explanation of and motivation for the method generally assume a lot of background; for example, what is meant by "separability", "system capacity", "manifold" and "manifold capacity", "random dichomoties". As such, it was not easy to understand all details of the method.

**Summary Of The Paper:**

The submission presents an extension to the representational analysis framework of Chung et al. (2018) in order to apply the framework to more standard computer vision tasks such such as multiway classification, with the goal of more precise comparisons between humans and neural networks.

**Summary Of The Review:**

The submission needs improvement in clarity to better convey its contributions to the neuroscience and computer vision communities.

---

> ### Author Response · Authors · 2022-11-17
> **Response**
>
> We are glad the reviewer finds our work applicable to computer vision tasks and allows for a more “precise comparison between humans and neural networks.” We have improved clarity of definitions in sections 2 and 3 in the updated version. We also plan to provide an additional glossary in the appendix of the final version of the paper upon acceptance. Below, we clarify some specific phrases that were asked and express any corresponding changes in the updated version.
>
> - In the context of our work,  “separability” is the existence of a linear hyperplane that is able to divide manifolds in their feature space. In the revised version, instead of “separability,” we will use the term “linear separability.” We revised the last sentence of the first paragraph in the introduction to help clarify this point:
> > “A long-standing hypothesis in visual neuroscience posits …, relating object recognition to the separation of manifolds by some linear hyperplane.”
> - “System capacity” was used in previous works (Cohen et al) in reference to Gardner’s perceptron capacity. We do recognize that this term is ambiguous as it has been used to discuss network capacity. In light of this, we will simply use “capacity” when referring to Gardner’s work.
> - “Manifold capacity” follows the definition in Chung et al and which we defined as “the maximum number of object manifolds that can be separated given a random dichotomy.” However, since many previous works that have employed MFTMA analyzed manifold capacity in the general case of dense labeling, we distinguish the sparse labeling case in our paper with the phrase “sparse manifold capacity.”
> - In the context of computational neuroscience and our paper, “manifold” refers to “object manifold” and is the “underlying representation of neural responses to a distinct object class.”
> - A random dichotomy is a random assignment of binary labels to the manifolds. We included this definition in the main text when dichotomy is first introduced:
> > “The theory quantifies a capacity load that describes the maximum number of points that can be linearly separated given a random dichotomy (a random assignment of binary labels to each manifold)”

---

### Official Review · Reviewer_pKPQ · 2022-11-03

**Confidence:** 4
**Clarity, Quality, Novelty And Reproducibility:** The novelty of this paper is limited …
**Correctness:** 3
**Technical Novelty And Significance:** 2
**Empirical Novelty And Significance:** Not applicable
**Recommendation:** 3

**Strength And Weaknesses:**

Strengths:

1.The proposed Sparse Replica Manifold analysis overcomes the limitation of size of neuroscience dataset and is close to real tasks in cognitive science.

2.The authors analyze the manifold capacity of both deep neural network manifolds and neural manifolds, making the conclusions more convincing.

Weaknesses:

1.The authors follow the work of Chung et al. (2018), but they don’t clearly claim the contributions of previous work.

2.The datasets this paper uses are relatively old and limited.

3.More comprehensive and thorough experiments are necessary to prove the conclusions.

Questions for the Authors

1.In chapter 3, the authors claims that sparse manifold capacity has been previously considered in Chung et al. (2018) and that they extend this work k by taking into account of heterogeneous geometries and correlation between manifolds. But in “RESULTS”, they don’t clearly claim the advances of Sparse Replica Manifold compared with previous works and the contributions of taking different Gaussian Radii and Dimensions into account.

2.The dataset this paper uses are from Majaj et al. (2015), Freeman et al. (2013) and VGG-16 trained by a CIFAR-100. More experiments on latest and large-scale datasets are needed.

3.In Figure 5, the change ranges of Dg in both neural dataset and the tendency of Rg and Deff is not so evident. More experiments are necessary to prove the conclusion.

4.The broader application of manifold analysis is meaningful; however, the scale of neuroscience dataset is larger recently. The total sequential computation time of Spares Replica Manifold Analysis has also no advantage. The influence of this method needs to prove again.

**Summary Of The Paper:**

This paper extends Chung et al. (2018), which proposes Sparse Replica Manifold analysis to estimate manifold capacity. The authors show that the application of sparse manifold capacity requires a smaller number of features and is faster compared to dense labeling. The authors also illustrate the effects of ambient dimension and manifold intrinsic dimension on sparse separability.

**Summary Of The Review:**

This paper is the following work of Chung et al. (2018) and many conclusions have been proposed before.

---

> ### Author Response · Authors · 2022-11-17
> **Response Part 1**
>
> We thank the reviewer for their valuable comments and suggestions. We are glad that the reviewer finds our conclusions are “convincing” through application of the Sparse Replica Manifold analysis on real neural data and artificial data. We address the reviewer’s concerns and questions below:
>
> **“1.In chapter 3, the authors claims that sparse manifold capacity has been previously considered in Chung et al. (2018) and that they extend this work k by taking into account of heterogeneous geometries and correlation between manifolds. But in “RESULTS”, they don’t clearly claim the advances of Sparse Replica Manifold compared with previous works and the contributions of taking different Gaussian Radii and Dimensions into account.”**
>
> We acknowledge that section 4 does not clearly claim the advances in Sparse Replica Manifold analysis compared to Chung et al. In Chung et al (PRX 2018), the theory for sparse label covers only a limited case of toy data, namely the shapes of the manifolds were assumed to be identical, and the distribution of manifold centroids were assumed to be random Gaussian. Hence while PRX 2018 (Chung et al) set up the stage for calculating the manifold capacity for simple cases as above, it wasn’t directly applicable due to the restrictive assumption (such as homogeneity and no correlations). Cohen et al (2020) took the first step towards making the manifold capacity theory applicable to real data, by taking the correlations between centroids into account, but it was limited to the capacity with dense labels. In this work, we fill in the gap of applying sparse capacity to real data. We provide the theorem necessary to compute capacity with sparse labels with application to real neural data that takes both center correlations and varying manifold shapes into account. To clarify this point, we added the following to the beginning of section 4:
>
> > “By relaxing the homogeneous and random correlations assumptions in Chung et al. (2018), our method estimates manifold capacity for real data in the one-vs-rest recognition limit, the relevant domain for machine learning and neuroscience. While previous works (Cohen et al., 2020; Stephenson et al., 2019) estimate manifold capacity in ANNs under the dense label regime, we demonstrate for the first time the application of manifold capacity in the neuroscience domain, enabled by sparse labeling. “
>
> **“2.The dataset this paper uses are from Majaj et al. (2015), Freeman et al. (2013) and VGG-16 trained by a CIFAR-100. More experiments on latest and large-scale datasets are needed.”**
>
> Thank you for pointing this out. We used these datasets (Majaj et al and Freeman et al) as they are benchmark datasets on Brainscore (Shrimpf et al, 2018 [1]) commonly used to evaluate artificial model consistency to the visual cortex (Sorcher et al, 2021[2]; Kong et al, 2021[3]; Anand et al, 2020[4]; Han et al, 2022[5]; Dapello et al, 2022[6]). Since our theory of sparse manifold capacity focuses on the one-vs-rest recognition regime, which is relevant to both neuroscience and machine learning, it enables a metric of comparison between humans and neural networks; hence, we believe it is reasonable to evaluate our theory on these datasets. We have, however, included an experiment on Imagenet-trained Resnet101 in the appendix section. We also show the results here https://imgur.com/gahsWOA. UPDATED: See also https://imgur.com/WSoX1j2 for results with Resnet50.
>
> [1] Schrimpf, Martin, Jonas Kubilius, Ha Hong, Najib J. Majaj, Rishi Rajalingham, Elias B. Issa, Kohitij Kar, Pouya Bashivan, Jonathan Prescott-Roy, Franziska Geiger, Kailyn Schmidt, Daniel L. K. Yamins, and James J. DiCarlo. Brain-score: Which artificial neural network for object recognition is most brain-like? bioRxiv preprint, 2018. URL https://www.biorxiv.org/content/
> 10.1101/407007v2
>
> [2] Ben Sorscher, Surya Ganguli, and Haim Sompolinsky. The geometry of concept learning. bioRxiv, 2021
>
> [3] Kong, N.C., Margalit, E., Gardner, J.L., & Norcia, A.M. Increasing neural network robustness improves match to macaque V1 eigenspectrum, spatial frequency preference and predictivity. PLoS Computational Biology, 18. 2021
>
> [4] Anand, Aditi, Sanchari Sen and Kaushik Roy. “Quantifying the Brain Predictivity of Artificial Neural Networks With Nonlinear Response Mapping.” Frontiers in Computational Neuroscience 15, 2020
>
> [5] Han, Yena, Tomaso A. Poggio and Brian Cheung. “System identification of neural systems: If we got it right, would we know?” 2022
>
> [6] Dapello, Joel, Kohitij Kar, Martin Schrimpf, Robert F. Geary, Michael Ferguson, David D. Cox and James J. DiCarlo. “Aligning Model and Macaque Inferior Temporal Cortex Representations Improves Model-to-Human Behavioral Alignment and Adversarial Robustness.” bioRxiv, 2022

---

> > ### Author Response · Authors · 2022-11-17
> > **Response part 2**
> >
> > **“3.In Figure 5, the change ranges of Dg in both neural dataset and the tendency of Rg and Deff is not so evident. More experiments are necessary to prove the conclusion.”**
> >
> > We thank the reviewer for pointing this out. While the tendency in geometric changes for neural data is not as clear and appears to differ from the observations in the artificial network, we emphasize the results in this paper is the first time manifold capacity is applied on real neural datasets which was previously restricted by the limitations in Chung et al., 2018 and Cohen et al., 2020. The conclusions made by Cohen et al were that neural networks exhibit decrease in manifold dimension and radius while increase in capacity. In the neural data, there is a decrease in manifold gaussian radius and effective radius but small changes in Gaussian dimension and effective dimension. The differences between the manifold geometries in the visual cortex and artificial models emphasizes that the manifold geometric measures could be used as a metric of comparison between artificial models and the human visual cortex. To elucidate this point, we add the following to the end of section 4.4
> >
> > > “...the manifold Gaussian dimension and effective dimension appear to exhibit the opposite behavior relative to the artificial network. These suggestive differences between the artificial and the biological brain points toward using manifold geometries as a metric to
> > finding and fitting an artificial brain model“
> >
> > **“4.The broader application of manifold analysis is meaningful; however, the scale of neuroscience dataset is larger recently. The total sequential computation time of Spares Replica Manifold Analysis has also no advantage. The influence of this method needs to prove again.”**
> >
> > We acknowledge that the recent scale of neuroscience data has increased, however, our work provides the technique to estimate capacity in a relevant setting (one vs rest) for neuroscience tasks that would be necessary regardless of the dataset size. Previous works, such as Chung et al and Cohen et al, while setting up the stage for calculating capacity, do not address the one vs rest setting that is most relevant in the neuroscience domain and to enable a comparison between neural networks and the human visual cortex. Most common tasks in neuroscience experiments require their subjects perform one-vs-rest classification such as the oddity task (where subjects find the odd object in a group) and delayed matching (where subjects pick the object in the same class to one they were previously shown). In addition, artificial neural networks use one-hot encodings as label classes and whose accuracy is evaluated on identifying one correct class versus the rest. Our work provides the method necessary to examine capacity in ways that are relevant for the neuroscience and machine learning communities. Sparse manifold capacity can be used as a measure of object recognition, towards studying the dynamics in neural data, and as a metric of similarity between the brain and an artificial model. Upon publication, the code will be made open sourced for the broader neuroscience community.

---

### Author Response · Authors · 2022-11-17
**Overall Respoonse**

We are grateful for the reviewers’ time spent reviewing this paper and their invaluable comments to improve our work. Our paper provides a method to estimate sparse manifold capacity in real neural data in a relevant domain for common neuroscience tasks such as object recognition. We hope that the theoretical and algorithmic contributions introduced in this work broadens the accessibility of the manifold capacity framework to the wider neuroscience community[*]. We are pleased that the reviewers find our work “valuable” and “beneficial to both computational neuroscience and machine learning communities.” In addition, we are glad the reviewers find our method offers “precise comparisons between humans and neural networks” through a sparse manifold capacity framework.

Here, we summarize some changes to our paper that we have incorporated from suggestions of the reviewer. Based on comments, we have made revisions to make the paper more self-contained by improving clarity in definitions. We thank the reviewer for pointing out unclear sentences, which we have improved. In the appendix section, we have included additional results that respond to questions from the reviewer. We responded to each reviewer’s individual questions below.

[*]: the analysis code will be released and open sourced upon acceptance

---

### Decision · Program_Chairs · 2023-01-20

**Decision:**

Reject

**Justification For Why Not Higher Score:**

The paper is not ready for publication. Needs better discussion of prior work to establish that there is sufficient novelty, and a more convincing use-case.

**Justification For Why Not Lower Score:**

NA

**Metareview: Summary, Strengths And Weaknesses:**

(a) The paper addresses the problem of calculating the capacity of object manifolds. Namely a set of manifolds, each corresponding to a particular object (e.g., the set of neural responses to this object). Prior work by Chung had presented a notion of capacity for such models (extending notions of capacity for a set of points, e.g., VC dimension). This work builds on Chung but focuses on the one-vs-all setting, and extends the results to that case.
(b) The extension to one-vs-all seems adequate for some cases of interest, and the application to neuronal data, and comparison to learned models is potentially of interest.
(c) One issue is that this is not a very big increment over Chung, and it would have been better to give a clearer exposition of Chung along with a clear description  of the difference (and which new technical tools were introduced). Another is that the neuronal data is somewhat outdated, and there is no clear conclusion or discussion regarding what this teaches us about neural coding.